# Learning Dense Object Descriptors from Multiple Views for Low-shot Category Generalization

**Stefan Stojanov, Anh Thai, Zixuan Huang, James M. Rehg**
Georgia Institute of Technology
{sstojanov, athai6, zixuanh, rehg}@gatech.edu

## Abstract

A hallmark of the deep learning era for computer vision is the successful use of large-scale labeled datasets to train feature representations. This has been done for tasks ranging from object recognition and semantic segmentation to optical flow estimation and novel view synthesis of 3D scenes. In this work, we aim to learn dense discriminative object representations for low-shot category recognition *without* requiring any category labels. To this end, we propose Deep Object Patch Encodings (DOPE), which can be trained from multiple views of object instances without any category or semantic object part labels. To train DOPE, we assume access to sparse depths, foreground masks and known cameras, to obtain pixel-level correspondences between views of an object, and use this to formulate a self-supervised learning task to learn discriminative object patches. We find that DOPE can directly be used for low-shot classification of novel categories using local-part matching, and is competitive with and outperforms supervised and self-supervised learning baselines. Code and data available at `https://github.com/rehg-lab/dope_selfsup`

## 1 Introduction

Achieving high accuracy at object category recognition generally requires deep models with millions of parameters [26, 13] and million-scale datasets [11, 37]. Alleviating this data requirement is a fundamental challenge for computer vision and has driven the development of low-shot [57, 65, 17, 76, 5] and self-supervised [43, 25, 4, 22] methods for learning object representations. Despite the increasing interest in low-shot and self-supervised learning, only a few works tackle the case where multiple unlabelled views of object instances are available for self-supervised learning of discriminative object representations [27, 32]. In addition, their primary focus is not on low-shot generalization. Using multi-view data to learn representations for low-shot categorization in a self-supervised manner is of practical interest, as such data can potentially be obtained from robots moving around and handling objects [19, 23] or by images and videos collected by humans using mobile devices [77, 53].

The goal of this paper is to address this gap by using multiple-views of many object instances for training, combined with some additional geometric information about the views, to learn representations that can be directly used for low-shot object category recognition. We assume access to sparse depth, known camera pose/calibration, and foreground segmentation during training, *with no access to category labels*. We use this information to formulate a self-supervised representation learning task. In practice, the additional data necessary for self-supervision can be obtained in controlled indoor settings using robotic arms equipped with RGB-D sensors or by postprocessing RGB-D sensor data and using forward kinematics to obtain the camera pose [19, 23]. Otherwise, in less constrained settings where the data is videos of objects captured using mobile devices, the data needed for self-supervision can be obtained from a Structure-from-Motion (SfM) and Multi-View

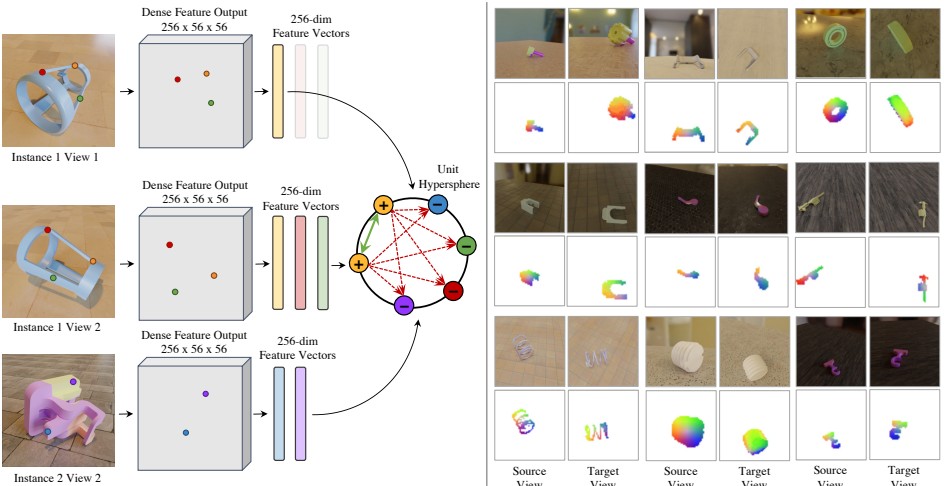

Figure 1: **Left:** Our contrastive learning formulation: We form positives between features at pixel locations of the same 3D point on the object surface, and negatives pixels from different points on the object surface. **Right:** For each feature corresponding to a point on the predicted segmentation mask in the source view we assign a color. For each point in the predicted segmentation for the target views, we assign the color of the highest cosine similarity feature from the source view. We see that our model learns dense object encodings which are consistent across many views of an object.

Stereo (MVS) pipeline such as COLMAP [55, 56]. Note that we do not assume access to any sort of category-level object or part annotations or any object attribute information for representation learning. At test time, our method only uses single-view RGB images.

To solve this self-supervised learning task, we propose Deep Object Patch Encodings (DOPE). DOPE uses pairs of object views to learn a dense, CNN-based local feature descriptor that will map a patch on the surface of the object that is visible in both views to the same feature vector, and different surface patches to different feature vectors (see Figure 1). We define an object patch as a local neighborhood on the 3D surface of a rigid object. Surprisingly, we find that such a representation, trained only on pairs of views of the *same* object instances, *generalizes across novel object categories* (illustrated in Figures 6 and 5). This is demonstrated by the ability to encode the same object parts, e.g. a motorcycle tire, or an airplane wing, with the same learned feature vector across multiple instances of a category. This generalization ability allows us to directly use the DOPE representation (trained only on local patch matches with no category labels) for low-shot object recognition.

DOPE is trained using a local contrastive learning loss which requires correspondences between views to form positive and negative pairs. To find such pairs, we use sparse depth map and camera information. Similar contrastive formulations have been used before to train instance-specific or class-specific dense object descriptors [19, 23] from RGB images for robotics applications, but did not address low-shot recognition. We hypothesize that learning dense surface patch similarity at the local feature level forces the model to learn to encode all visible object parts in two views, and find that such a representation is useful for low-shot object categorization.

We find that DOPE is competitive and in some cases outperforms both self-supervised methods [27] and fully supervised low-shot methods [68, 62, 76] on the synthetic ModelNet [71], ShapeNet [3] and the real-world CO3D [53] datasets. We further find that DOPE can learn a representation using objects from the large, uncurated 3D object dataset ABC [35], which directly transfers to low-shot category recognition. In summary, we make the following contributions:

- Deep Object Patch Encodings (DOPE), a self-supervised learning approach based on learning object shape correspondences, generalizes to low-shot object categorization.

- The first study of how a large, unstructured object instance dataset like ABC [35] can be used effectively to learn representations for low-shot category recognition from images.

- Multiple datasets with high photorealism rendered from existing 3D assets that will enable future research in learning representations using multi-view information.

## 2  Related Work

### 2.1  Low-shot Learning for Object Recognition

The goal of low-shot learning is to build models that can learn how to classify images based on a few labeled samples. These are typically trained on a large base class dataset, and their low-shot generalization ability is evaluated on many low-shot episodes with a few labeled training images and testing queries from novel, unseen classes. Such models can be categorized based on how the base class data is used, and whether the model's parameters are updated during the low-shot phase. Based on base class data, the *meta learning*-based methods [57, 30, 72, 17] sample low-shot episodes from the base classes to train under the same setting as during evaluation. In contrast, the *whole-classification*-based techniques [68, 62] train the backbone feature representation as a classifier on the base classes, whereas [72, 8, 20, 41] further fine-tune the representation with a meta-learning procedure. Based on whether the model parameters are updated, *optimization*-based techniques [54, 17, 18] aim to learn a representation that can be fine-tuned using a few samples while *metric learning*-based techniques [68, 72, 8, 62] learn a discriminative metric space that will directly transfer to low-shot episodes of novel classes. Our self-supervised representations do not require any labels during training, and like metric learning techniques, can be directly applied to novel classes.

### 2.2  Self-Supervised Learning

Self-supervised learning aims to formulate tasks that exploit the underlying structure of the data rather than category labels to train deep network-based representations suitable for downstream tasks.

**Self-Supervised Learning for Object Classification**    Early works formulated proxy tasks such as colorization [78], predicting relative positions of patches [12, 43] or predicting rotations [21]. More recently, methods based on instance-level comparisons have led to significant performance improvements. These are trained to map multiple views (usually multiple augmented versions of an image) as the same feature vector under some similarity measure, and different images as different vectors. SimCLR [4] and MoCLR [61] consider augmentations of the same underlying image as positives and other images in the minibatch as negatives, whereas the MoCo-based [6, 25] approaches use a continuously updating queue of negative features. The methods presented in [61, 6, 25, 1, 2] all use a second momentum-updated encoder rather than using the same backbone to encode multiple views of an instance. Our proposed approach makes use of a momentum encoder as well. Further, some works like [22, 7, 1, 2] achieve superior downstream performance for classification without requiring any explicit negatives, while others [24] obtain high performance by revisiting image reconstruction as a proxy task using vision transformers [13].

**Self-Supervised Learning for Dense Vision Tasks**    The main ideas from self-supervised contrastive learning for classification have been extended to learn local, dense, pixel-level representations for tasks like semantic segmentation, keypoint, and object part learning. For segmentation and detection, [75, 67, 73, 45] apply contrastive losses at the pixel level, treating features corresponding to individual pixels or pooled regions as positives or negatives, or alternatively use clustering-based [79] losses to fine-tune visual transformers. For keypoint learning, Cheng et al. [10] use a dense contrastive strategy, while Novotny et al. [44] propose a probabilistic matching loss. In addition, SCOPS [31] is a self-supervised technique for object part segmentation of fine-grained categories. All of these prior works are trained using pairs of images that have undergone 2D geometric transforms, while we use images corresponding to different camera positions and focus on low-shot categorization.

**Self-Supervised Learning using 3D Geometry and Multi-View Images**    Self-supervised learning has been studied in settings where multi-view RGB-D information (images and in-camera dense 3D Point clouds) is available [29, 74, 14, 48, 36, 19]. First, [29, 15, 38, 36, 48] use pixel-level correspondences between views of a scene, where a positive pair of pixels represents the same 3D point in the world projected in each image. Such pixel-level correspondences have been used to align [1] image and depth-based scene representations [29, 15, 38] for point cloud registration [15] and scene segmentation [38, 28], or to learn 3D CNN-based scene representations from multiple views [36, 48]

---

[1]Here by align we mean make corresponding features between the 2D and 3D representations have high similarity.

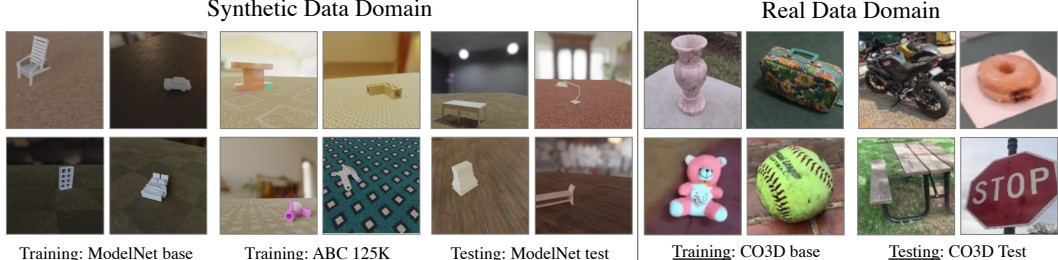

Synthetic Data Domain      Real Data Domain

Training: ModelNet base    Training: ABC 125K    Testing: ModelNet test    Training: CO3D base    Testing: CO3D Test

Figure 2: **Left:** Synthetic data rendered from ModelNet and ABC (first and second columns) for self-supervised learning of representations and low-shot testing data from ModelNet (third column). **Right:** Real data from CO3D for self-supervised representation training and low-shot testing.

for self supervised object tracking and few-shot style/shape learning. Florence et al. [19] use RGB images to learn dense object descriptors for robotics applications. In comparison, our goal is to use correspondences between multi-view RGB images of objects to learn representations for low-shot object classification. Other works [74, 28] directly train on 3D point clouds. Last, assuming only access to multiple RGB views, [32] uses an encoder/decoder architecture to predict images of different viewpoints from a single image, whereas VISPE [27] formulates an instance discrimination task. While we share with these works the use of self-supervised representation learning from multiple RGB views for object recognition, we focus on low-shot object classification. We show that our approach outperforms VISPE [27]-like instance-level global representations on multiple datasets.

### 2.3 Object Classification on Synthetic Datasets

Object recognition has been studied using 3D datasets such as ModelNet40 [71] converted into point clouds [50, 52, 9, 69], voxels [51, 71], or rendered into images [60, 51, 16, 42, 27, 70, 32]. We use photorealistic rendering to generate data, and conduct the first study to investigate the use of the large unstructured ABC [35] dataset for representation learning in low-shot object recognition.

## 3 Learning Deep Object Patch Encodings - DOPE

We aim to learn a representation based solely on multi-view images of object instances that can directly generalize to low-shot object category recognition tasks. For training, we assume access to relative pose between views of object instances, foreground/background segmentation, and sparse depth. However, for low-shot generalization at test time, we only assume access to single RGB images. We do not make any assumptions about the semantic structure of the data for representation learning, or about the availability of any semantic labels such as explicit parts or attribute information.

In this section, we describe our approach for contrastive learning of dense object descriptors from multiple views that can generalize to object categories, and present a simple local nearest neighbor-based method to utilize this learned representation for low-shot classification.

### 3.1 Local Object Representation Learning

Assume we have a dataset $\mathcal{D}$ of object instances $o_i$, where each object has $N$ calibrated views with masks and known camera intrinsics and extrinsics. Denote the two views of one object as $\mathbf{v}_1^{o_1}$ and $\mathbf{v}_2^{o_1}$. If these two viewpoints are sufficiently close, there will be points on the surface of the object, e.g. one leg of a chair, that will be visible in both views. This means that the same point on the object surface, will get projected to pixel coordinates $(u_1, v_1)$ in $\mathbf{v}^{o_1}$ and $(u_2, v_2)$ in $\mathbf{v}_2^{o_1}$. Let $f : \mathbb{R}^{H \times W \times 3} \to \mathbb{R}^{h_f \times w_f \times D}$ be a learnable feature map that maps RGB images to a 2D feature grid with $D$ channels and $h_f$ and $w_f$ height and width. Let $\tilde{u}_k$ be the location in the lower spatial dimensional feature output corresponding to $u_k$. Our goal is to learn $f$ so that the feature vectors for the corresponding pixel locations $\mathbf{z}_1^{o_1} = f(\mathbf{v}_1^{o_1})[\tilde{u}_1, \tilde{v}_1, :]$ and $\mathbf{z}_2^{o_1} = f(\mathbf{v}_2^{o_1})[\tilde{u}_2, \tilde{v}_2, :]$ are so that $\mathbf{sim}(\mathbf{z}_1^{o_1}, \mathbf{z}_2^{o_1}) = 1$, where $\mathbf{sim}$ is some similarity metric, in our case normalized dot product. Conversely, we want $\mathbf{sim}(\mathbf{z}_1^{o_1}, \bar{\mathbf{z}}_2^{o_1}) = 0$, where $\bar{\mathbf{z}}_2^{o_1}$ is a feature from a pixel projected from the object surface in $\mathbf{v}_2^{o_1}$ that is not in correspondence. Further, for a feature $\bar{\mathbf{z}}_2^{o_2}$ extracted from some different object $o_2$, we want $\mathbf{sim}(\mathbf{z}_1^{o_1}, \bar{\mathbf{z}}_2^{o_2}) = 0$. Note that $\mathbf{z} \in \mathbb{R}^D$. In other words, following the chair leg

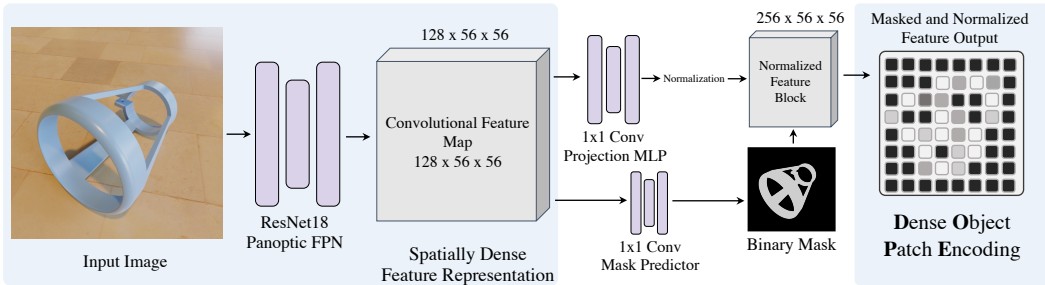

Figure 3: Our architecture consists of a ResNet18 [26] followed by a Panoptic Feature Pyramid network [33] to output a 2D grid of convolutional features. We use an additional branch from these features to predict a binary mask. After applying a series of 1x1 convolution layers, we perform channel-wise unit vector normalization and masking with the binary mask to obtain the final feature output we use for contrastive training and at test time

example, we want $f$ to produce the same feature in the 2D feature grid for a point on the chair leg in any viewpoint where that chair leg is visible (Please refer to Fig. 1 for illustration).

To learn such a representation, we sample a batch of objects $\mathcal{B}$, and for each object $o_k \in \mathcal{B}$ we sample an image pair $\mathbf{v}_1^{o_k}$ and $\mathbf{v}_2^{o_k}$. For each view pair, we sample a set $\mathcal{C}$ of $n$ pixels on the object surface in $\mathbf{v}_1^{o_k}$ using farthest point sampling and find the corresponding pixels in $\mathbf{v}_2^{o_k}$ [2]. For each pixel in $\mathcal{C}$ we therefore obtain the feature vector $\mathbf{z}_1^{o_k}$ and the corresponding $\mathbf{z}_2^{o_k}$ whose similarity $\mathbf{sim}(\mathbf{z}_1^{o_k}, \mathbf{z}_2^{o_k})$ we want to maximize. Concurrently, we want to minimize: (1) $\mathbf{sim}(\mathbf{z}_1^{o_k}, \bar{\mathbf{z}}_2^{o_k})$ for any feature $\bar{\mathbf{z}}_2^{o_k}$ from a non-corresponding pixel on the object and (2) $\mathbf{sim}(\mathbf{z}_1^{o_k}, \bar{\mathbf{z}}_2^{o_j})$ for any feature $\bar{\mathbf{z}}_2^{o_j}$ from some other object where $o_j \in \mathcal{B}$, where $j \neq k$. To achieve this we minimize a normalized and temperature-scaled cross entropy loss [58, 6] for each pair of positive correspondences

$$\ell = -\log \frac{\exp\left(\mathbf{sim}(\mathbf{z}_1^{o_k}, \mathbf{z}_2^{o_k})/\tau\right)}{\sum_{\bar{\mathbf{z}}_2^{o_k}} \exp\left(\mathbf{sim}(\mathbf{z}_1^{o_k}, \bar{\mathbf{z}}_2^{o_k})\right)/\tau + \sum_{j, j \neq k} \sum_{\bar{\mathbf{z}}_2^{o_j}} \exp\left(\mathbf{sim}(\mathbf{z}_1^{o_k}, \bar{\mathbf{z}}_2^{o_j})\right)/\tau}, \quad (1)$$

where $\tau$ is a temperature parameter controlling how peaky the softmax output is. This formulation is commonly used in contrastive learning. We apply this loss for all sampled points in each image pair and refer to it as $\mathcal{L}_{\text{corr}} = \sum_{\mathcal{C}} \ell$.

To avoid learning to encode the background, we learn an additional module $h : \mathbb{R}^{h_f \times w_f \times D} \to \mathbb{R}^{h_f \times w_f}$ to predict binary masks $\widehat{\mathbf{m}}$ which we train with binary cross entropy $\mathcal{L}_{\text{mask}} = BCE(\widehat{\mathbf{m}}, \mathbf{m})$. We then apply this mask to the final output of $f$ (refer to Fig. 3) to mask out the background. We optimize the sum of both losses with equal weight. Fig. 1 contains qualitative examples of how our proposed approach matches object shape across viewpoints.

## 3.2 Low-Shot Object Recognition using DOPE

Given that we have the means to encode the same object part from different images (see Figures 5 and 6), we present a simple nearest neighbor-based method to make use of this representation for low-shot category recognition. This method is outlined in Figure 4, and we give a formal description next. Let $\mathbf{Z}_q \in \mathbb{R}^{D \times h_f \times w_f}$ be a feature map of a query image that we wish to classify given a labeled support set of $N$ images. Our goal is to find the image from the support set which is the most similar to the query and assign its label to the query image.

We first sample $k$ pixel locations from the predicted foreground of the query image using farthest point sampling and extract their corresponding feature vectors $\mathbf{z}_q^1, \ldots, \mathbf{z}_q^k$. We then extract complete feature grids for each of the shot images and flatten them into matrices $\mathbf{Z}_s^1, \ldots, \mathbf{Z}_s^N \in \mathbb{R}^{(h_f \cdot w_f) \times D}$. Note that the $\mathbf{z}$ vectors and the columns of the $\mathbf{Z}$ matrices are unit normalized. The product $\mathbf{s}_q^i = \mathbf{Z}_s \mathbf{z}_q^i$ where $\mathbf{s}_q^i \in \mathbb{R}^{h_f \cdot w_f}$ is the similarity of the query feature vector $\mathbf{z_q}$ at the $i$-th sampled pixel location and all features $\mathbf{Z}_s^n$ from the $n$-th support image. We compute $\mathbf{s}_q^i$ for all $k$ points in the query image, and use $\sum_{i=1}^{k} \max(\mathbf{s}_q^i)$ as the score for that query/support image pairs. We take the label from the

---

[2]We obtain these correspondences using the segmentation masks, known camera intrinsics and extrinsics and depths. We also use the depth information to avoid sampling points that are occluded.

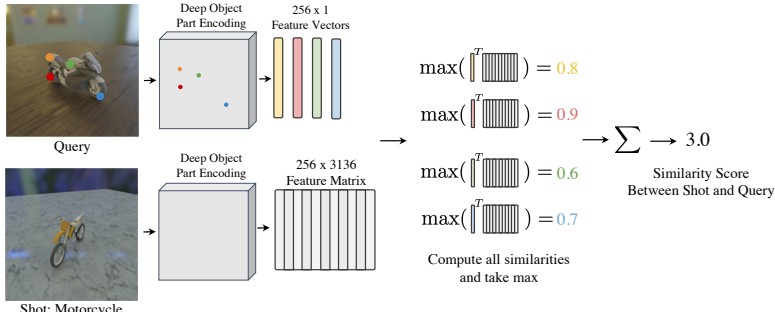

Figure 4: To determine a similarity score between a labelled shot and query image, for each query image, we first use farthest point sampling to randomly choose $k$ pixels on the object surface, based on a predicted segmentation mask (in this example $k = 4$). We then extract features corresponding to those points from the query, and a full feature grid for the shot. After computing the cosine similarity between all query and each shot features, we find the maximum similarity score between each shot and all query features. We sum these scores for the final similarity score.

support image with the highest score and assign it to the query. When there are multiple shot images per category available, we perform a 1-nearest neighbor classification.

## 4  Implementation Details

The backbone of DOPE is based on a ResNet18 [26] with a Panoptic Feature Pyramid network [33], taking as input a $3 \times 224 \times 224$ image and outputting a $256 \times 56 \times 56$ 2D feature block. This backbone formulation follows the approach in VADeR [46]. The FPN output is then sent through a set of 1x1 convolution layers with dimension 128-1024-1024-256. During training, we use $n = 32$ pixel locations on the object surface for each image pair, and use a batch size of 256 view pairs. DOPE is trained using AdamW [39] and an initial learning rate of $10^{-4}$ and weight decay of $10^{-2}$, for 3000 epochs, using a cosine learning rate annealing schedule. For each view pair, one image is sent to a momentum-encoder backbone as in [25]. At test time, we use $k = 20$ pixel locations for our local nearest neighbor-based classification approach. During training, we use the known segmentation mask to randomly remove the background and perform flipping, color jittering and additional photometric augmentation, building on [19]. We empirically found that this random background removal significantly improves the model's generalization ability, forcing it to learn to encode cues from the foreground object. The self-supervised baselines and low-shot baselines in the Experiments (Section 5) are trained using the same background masking strategy we use for DOPE to ensure a fair comparison. Further implementation details are provided in the Appendix.

## 5  Experiments

In this section, we demonstrate DOPE's low-shot generalization ability of in comparison to both supervised algorithms and strong self-supervised baselines on synthetic and real datasets. A surprising finding is that our model outperforms few-shot baselines trained on category labels for ModelNet40, and delivers comparable performance for CO3D. For a thorough comparison, all models are evaluated over five different randomly chosen validation and testing splits, where each validation split is used to find the best checkpoint for each test set. In all our experiments, parentheses show confidence intervals based on 2.5K low-shot episodes. We include further results and ablation studies in the Appendix.

### 5.1  Datasets

We use Blender [49] with the Cycles ray-tracing engine to create synthetic data, rendering 20 views per object with varying camera distance, azimuth, and elevation. Objects are on top of a plane with a PBR material and illuminated using image-based lighting from an HDRI environment map. These assets were collected from PolyHaven [47].

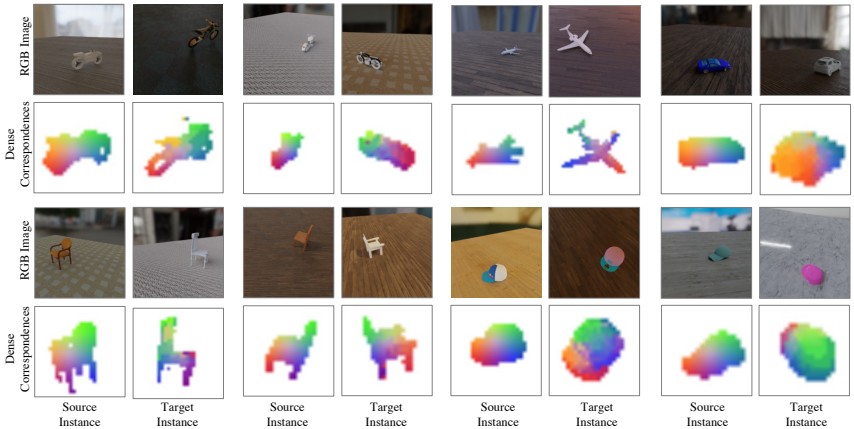

Figure 5: Category-level dense correspondences for ShapeNet [3] emerge from training on the ABC [35] instance dataset. For each feature corresponding to a point on the predicted segmentation mask in the source view we assign a color. For each point in the predicted segmentation for the target views, we assign the color of the highest cosine similarity feature from the source image. We see that our model learns dense object descriptors which transfer across instances of a category. We zoom into the object for the dense visualization images to improve visibility.

Following [32, 27, 59] and convention in low-shot learning we partition ModelNet [71], ShapeNet [3] and CO3D [53] into disjoint sets of base classes—used for representation training, validation classes—used for early stopping and hyperparameter tuning, and test classes—used to test low-shot category recognition. Further, we use the 3D object instance dataset ABC [35] to investigate whether a large and diverse set of object instances without any underlying category structure can be used to learn representations useful for low-shot category generalization. For illustration see Figure 2. Following are additional dataset details:

- **ModelNet40-LS**[71] is a dataset of 3D object categories split into disjoint sets of 20 training, 10 validation and 10 test categories. We use the 20 base classes with labels for training the supervised models, and without labels for the self-supervised models.
- **ShapeNet50-LS**[3] is a dataset of 3D object categories split into 25 training, 10 validation and 20 test categories. We use the 20 base classes with labels for training the supervised models, and without labels for the self-supervised models.
- **ABC**[35] is a dataset of 3D object instances from 3D printing online repositories without any category structure. We randomly select a set of 115K instances from this dataset for training and directly evaluate transfer to the ModelNet and ShapeNet low-shot test sets.
- **CO3D-LS**[53] (Common Objects in 3D) is a dataset which consists of 20K crowd-sourced videos of objects from 51 categories, where the videos are collected by a person moving around an object using a mobile device. We split CO3D into 31 base, 10 validation and 10 test classes. The dataset is post-processed using COLMAP [56, 55] to estimate the cameras and reconstruct the scene and object geometry. It also includes object segmentation masks estimated using PointRend [34].

## 5.2   Baseline Methods

We provide a brief overview of the supervised and self-supervised baseline algorithms we evaluate against. All baselines are implemented using the same ResNet18[26] backbone and are tuned to our setting (details are included in the Appendix).

- **SimpleShot**[68] is a simple and competitive low-shot baseline. A CNN-based backbone is trained with cross-entropy on the base classes and tested with nearest class mean classification.
- **RFS**[62] is a strong but simple baseline, trained with cross-entropy on the base classes, and a logistic regression-based classifier trained on the support data in each low-shot episode.
- **FEAT**[76] finetunes a transformer-based [64] set-to-set function on top of a cross-entropy-trained backbone. Methods that outperform FEAT on miniImageNet like COSOC [40] do so by eliminating learning classification by focusing on the background. As there is no correlation between classes and backgrounds in our synthetic setting, FEAT represents the state of the art in low-shot learning in our synthetic data comparisons.

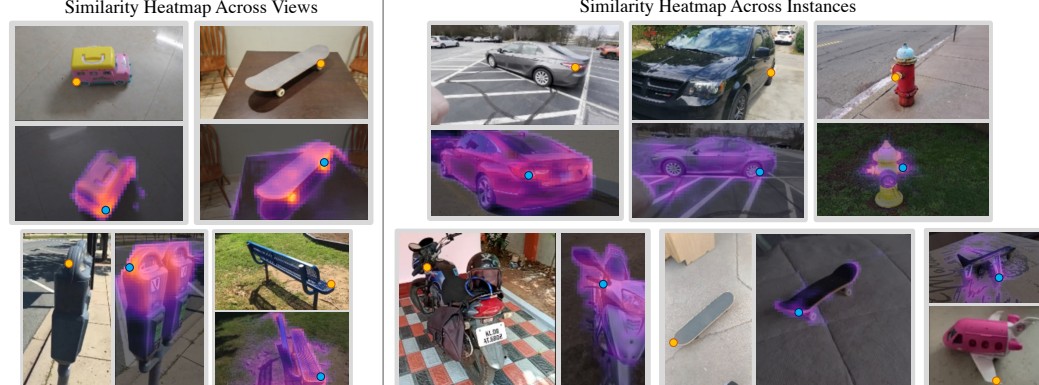

Figure 6: Similarity heatmap for the query point (yellow) **Left:** across different views of the same object and **Right:** across different object instances from CO3D. The blue point on the heatmap-overlaid image indicates the predicted correspondence of the query point. Note that these objects are from validation or test categories, which are different from the training categories.

Table 1: Results on image-only low-shot recognition on the **ModelNet40-LS**. DOPE trained on ABC outperforms supervised baselines and the global self supervised learning approaches. Dashed line separates supervised and self-supervised methods.

| Episode Setup → | 5-way | | 10-way | |
|---|---|---|---|---|
| | 1-shot | 5-shot | 1-shot | 5-shot |
| SimpleShot [68] | 56.55 ±0.42 | 69.87 ±0.32 | 41.27 ±0.24 | 54.84 ±0.18 |
| RFS [62] | 57.31 ±0.40 | 73.77 ±0.33 | 42.22 ±0.34 | 59.97 ±0.18 |
| FEAT [76] | 57.46 ±0.39 | 71.73 ±0.32 | 41.72 ±0.24 | 57.84 ±0.18 |
| SupMoCo [41] | 55.32 ±0.40 | 71.82 ±0.33 | 39.87 ±0.23 | 57.15 ±0.17 |
| VISPE [27] | 56.27 ±0.44 | 67.76 ±0.35 | 40.41 ±0.26 | 51.97 ±0.18 |
| VISPE++- SimSiam [7] | 53.83 ±0.29 | 68.75 ±0.25 | 39.84 ±0.17 | 54.34 ±0.13 |
| VISPE++ - MoCoV2 [25] | 57.05 ±0.42 | 71.81 ±0.35 | 43.23 ±0.25 | 58.68 ±0.18 |
| DOPE (ours) | 57.51 ±0.44 | 70.44 ±0.36 | 42.73 ±0.26 | 55.52 ±0.19 |
| VISPE++ - SimSiam [7] - ABC | 60.24 ±0.28 | **76.55 ±0.22** | 47.02 ±0.18 | **64.51 ±0.13** |
| VISPE++ - MoCoV2 [25] - ABC | 61.07 ±0.41 | 75.96 ±0.32 | 47.67 ±0.25 | 63.27 ±0.18 |
| DOPE (ours) - ABC | **62.76 ±0.43** | 76.86 ±0.33 | **49.39 ±0.26** | 64.77 ±0.18 |
| DOPE (ours) - ABC - [66] pred. mask | 57.63 ±0.40 | 71.62 ±0.31 | 43.43 ±0.24 | 57.72 ±0.18 |

- **SupMoCo**[41] uses a labeled base-class dataset to perform contrastive training, and trains a cosine-classifier [20] for each low-shot episode. We finetune a logistic regression-based classifier as in RFS due to the smaller episode sizes in our setting, upon consulting the authors. We found that it results in improved performance. SupMoCo is the state of the art for low-shot learning on MetaDataset [63].

- **VISPE**[27] is a self-supervised baseline that uses a temperature-scaled cross-entropy to learn global features so that two views of the same object are positives and two views of different objects are negatives. We use a nearest class mean classifier in the learned global feature space for low-shot classification as we found that it gives the highest performance.

- **VISPE++** We improve on the original VISPE implementation by applying more recent contrastive learning techniques based on either MoCoV2 [6] or SimSiam [7], resulting in stronger baselines. We use a 1-nearest neighbor classifier in the learned global feature space for low-shot classification as we found that it gives the highest performance.

## 5.3 DOPE trained on ABC outperforms supervised and self-supervised baselines on ModelNet and ShapeNet

We evaluate DOPE's ability to directly perform low-shot classification without any category supervision during representation learning on synthetic data, and demonstrate its improved low-shot generalization ability over both supervised and self-supervised baselines when trained on the ABC dataset. The results are presented in Table 1 and Table 2. We observe that when trained on ABC without any category labels, DOPE outperforms the self-supervised baselines. Note that both the SimSiam [7] and MoCoV2 [25] versions of VISPE++ and our proposed DOPE outperform supervised baselines by a large margin when trained on ABC.

Table 2: Results on image-only low-shot recognition on the **ShapeNet-LS**. DOPE trained on ABC outperforms supervised baselines and outperforms the global self supervised learning approaches. Dashed line separates supervised and self-supervised methods.

| | 5-way | | 10-way | |
|---|---|---|---|---|
| Episode Setup → | 1-shot | 5-shot | 1-shot | 5-shot |
| SimpleShot [68] | 58.07 ±0.45 | 70.06 ±0.39 | 43.00 ±0.29 | 55.96 ±0.25 |
| RFS [62] | 57.93 ±0.46 | 73.23 ±0.37 | 43.08 ±0.28 | 59.71 ±0.25 |
| FEAT [76] | 57.83 ±0.45 | 72.41 ±0.37 | 42.92 ±0.29 | 58.95 ±0.25 |
| VISPE | 57.69 ±0.47 | 68.65 ±0.39 | 42.43 ±0.29 | 54.00 ±0.24 |
| VISPE++ - SimSiam [7] | 53.22 ±0.30 | 66.75 ±0.31 | 39.12 ±0.19 | 52.25 ±0.17 |
| VISPE++ - MoCoV2 [25] | 55.83 ±0.42 | 69.11 ±0.36 | 41.14 ±0.28 | 54.86 ±0.25 |
| DOPE (ours) | 58.64 ±0.48 | 70.43 ±0.38 | 44.26 ±0.30 | 56.07 ±0.26 |
| VISPE++ - SimSiam [7] - ABC | 58.48 ±0.32 | 72.19 ±0.34 | 44.59 ±0.20 | 59.17 ±0.18 |
| VISPE++ - MoCoV2 [25] - ABC | 58.99 ±0.46 | 72.26 ±0.39 | 44.96 ±0.30 | 59.02 ±0.26 |
| DOPE (ours) - ABC | **62.00 ±0.48** | **73.55 ±0.38** | **47.93 ±0.31** | **60.72 ±0.27** |

Table 3: Results on image-only low-shot recognition on **CO3D-LS**. DOPE outperforms the global self supervised learning-only approaches. Dashed line separates supervised and self-supervised methods.

| | 5-way | | 10-way | |
|---|---|---|---|---|
| Episode Setup → | 1-shot | 5-shot | 1-shot | 5-shot |
| SimpleShot [68] | 62.44 ±0.41 | 79.18 ±0.29 | 49.43 ±0.26 | 68.32 ±0.17 |
| RFS [62] | 62.59 ±0.41 | **81.50 ±0.26** | 50.16 ±0.26 | **71.89 ±0.16** |
| FEAT [76] | **64.48 ±0.43** | 79.04 ±0.29 | **51.58 ±0.26** | 69.66 ±0.16 |
| SupMoCo [41] | 62.34 ±0.43 | 78.72 ±0.44 | 48.79 ±0.27 | 67.84 ±0.16 |
| VISPE | 54.68 ±0.43 | 70.12 ±0.35 | 40.85 ±0.25 | 56.46 ±0.20 |
| VISPE++ (SimSiam) | 56.02 ±0.27 | 74.29 ±0.21 | 43.45 ±0.17 | 62.88 ±0.12 |
| VISPE++ (MoCoV2) | 60.25 ±0.41 | 76.30 ±0.30 | 47.09 ±0.26 | 64.98 ±0.18 |
| DOPE (ours) | 61.77 ±0.44 | 75.16 ±0.32 | 48.07 ±0.27 | 62.97 ±0.17 |

We also investigate removing the assumption of known foreground masks for training DOPE, by extracting foreground masks for the ABC dataset using the off-the-shelf unsupervised instance segmentation method FreeSOLO [66]. The result is presented in the bottom row of Table 1. We observe that even when the assumption of known foreground masks is removed, DOPE's performance drops but remains competitive with supervised baselines when trained on ABC.

## 5.4 DOPE trained CO3D outperforms other self-supervised baselines

We also evaluate DOPE's low-shot generalization ability on real data and observe that it has improved low-shot generalization ability over other self-supervised baselines. The results are presented in Table 3. We observe that DOPE can successfully be trained on real data with estimated scene geometry and that it outperforms self-supervised learning baselines that do not perform any local feature learning in the 1-shot scenario by up to 1.5%-points.

## 5.5 DOPE Qualitative Results

We present qualitative results for dense object correspondences on ShapeNet [3] in Figure 5 and for CO3D [53] in Figure 6. DOPE learns to map the same object parts across different views of the same instance and generalizes further to different object instances in different viewpoints (the front and back legs of the chairs are correctly matched in Figure 5 and the rear light of the cars in Figure 6). Note that for CO3D the local representations were trained on a different set of object categories, and were trained on the ABC dataset for ShapeNet. Further, we observe an intriguing property that the model is uncertain for object parts that are similar, as shown in the similarity heatmap in Figure 6 (e.g. the wheels of the skateboard). This highlights the ability of DOPE to successfully encode local features of objects and generalize to unseen data.

## 5.6 Negative Sampling Strategy

We present an ablation study in Table 4 to understand the effect of how negatives are sampled on low-shot classification performance. Recall from the left side of Figure 1 and Section 3.1 that while positives can only be obtained by comparing corresponding pixels in two views of one object,

Table 4: Ablation study over different strategies for obtaining negative samples for contrastive training. We observe that is essential to sample from the second object view and from other object instances in the batch to train the best representation. All models are trained on ABC and evaluated on ModelNet40-LS.

| Episode Setup → | 5-way | | 10-way | |
|---|---|---|---|---|
| | 1-shot | 5-shot | 1-shot | 5-shot |
| DOPE - ABC - 2nd view + other objects | **62.76** $\pm 0.43$ | **76.86** $\pm 0.33$ | **49.39** $\pm 0.26$ | **64.77** $\pm 0.18$ |
| DOPE - ABC - other objects only | 61.51 $\pm 0.42$ | 75.54 $\pm 0.32$ | 47.95 $\pm 0.26$ | 62.65 $\pm 0.18$ |
| DOPE - ABC - 2nd view only | 43.50 $\pm 0.39$ | 56.58 $\pm 0.28$ | 30.58 $\pm 0.25$ | 39.06 $\pm 0.17$ |

Table 5: We investigate the tradeoff between using two views of an object corresponding to two different camera positions in the 3D world against two "views" generated by augmenting a single image twice. We find that using two physically different viewpoints during training has a high impact on generalization ability.

| Episode Setup → | 5-way | | 10-way | |
|---|---|---|---|---|
| | 1-shot | 5-shot | 1-shot | 5-shot |
| DOPE - ABC - multi-view augmented image pairs | **62.76** $\pm 0.43$ | **76.86** $\pm 0.33$ | **49.39** $\pm 0.26$ | **64.77** $\pm 0.18$ |
| DOPE - ABC - single-view augmented image pairs | 38.36 $\pm 0.43$ | 51.68 $\pm 0.33$ | 25.78 $\pm 0.26$ | 37.19 $\pm 0.18$ |

negatives can come either from non-corresponding points in the second view of an object or from other objects. We observe that it is essential to both sample negatives from the second view and from the other objects in the batch to obtain the best representation.

### 5.7 Training with Multiple 3D Views of Objects is Essential

While prior dense contrastive learning works [46, 67, 73] generate training pairs by augmenting a single 2D image of an object, our method is trained using two images corresponding to two different camera placements in 3D space. To understand the importance of different 3D viewpoints of an object, we train DOPE using the same loss but generate image pairs by applying two augmentations of the same image instead (see Table 5). We find that augmenting the same image to generate view pairs is insufficient for low-shot generalization.

## 6 Limitations and Discussion

The main limitation of our proposed approach is its requirement of estimated scene geometry in order to derive pixel-level correspondences between multiple views of object instances. While this information allows us to train better self-supervised representations that do not require category labels, it comes at a computational cost when applied to real datasets. Future work may include developing means to apply contrastive learning at the local level but without requiring explicit pixel-level correspondences, potentially making use of the temporal contiguity of videos.

**Negative Societal Impact** Contrastive training generally requires long training runs with multiple GPUs, which has potential for negative environmental impact. This may be addressed in the future through improvements in chip design and optimization of deep models.

## 7 Conclusion

Progress in few-shot and self-supervised learning is essential for overcoming the current requirements for large labeled datasets. This paper presents a self-supervised method that learns from multiple views of object instances and can recognize novel categories based on a few labels. Our findings are validated on both synthetic and real datasets. Qualitatively, we observe that our proposed approach can match semantic elements on the object surface (e.g. tires, chair legs) in a way that generalizes to novel object categories. Developing means to perform local contrastive learning without explicit multi-view pixel-level correspondences is an exciting direction for future work.

## 8 Acknowledgement

This work was supported in part by NIH R01HD104624-01A1, NIH R01MH114999, NSF OIA2033413, and a gift from Facebook. We thank Miao Liu and Wenqi Jia for their helpful feedback and discussion.

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
