# Appendix: Learning Dense Object Descriptors from Multiple Views for Low-shot Category Generalization

**Stefan Stojanov, Anh Thai, Zixuan Huang, James M. Rehg**
Georgia Institute of Technology
{sstojanov, athai6, zixuanh, rehg}@gatech.edu

## A    Appendix Overview

This appendix is structured as follows: We first present an ablation study for our model in Section B; In Section C we provide additional qualitative results on the CO3D [20] dataset; In Section D we provide additional details about the datasets used in our experiments and their licenses; In Section E we provide details on the baselines we use, their implementations and hyperparameters; In Section F we describe the compute resources used in our research.

## B    Ablation Study

We present empirical results for ablating different elements of our model. Using the ModelNet [26] dataset, we train models without randomly removing the background of the input as a data augmentation step during training (denoted as DOPE w/o random background remove) and without predicting the object mask during training and multiplying it with the local feature encoding (denoted as DOPE w/o mask prediction). The results on the first ModelNet validation set are presented in Table 1. We observe that removing either of these two elements significantly reduces the performance of our model. The reduction in performance because of not randomly removing the backgrounds potentially indicates that without this augmentation, our model uses background texture/geometry information to learn features that do not generalize across instances.

Table 1: Ablation study over two elements of our proposed approach: without randomly masking the foreground objects in input images during training, and without predicting the object mask and multiplying it with the local feature encoding. We observe significant reductions in performance in both cases. Evaluation is done on the ModelNet validation classes.

|  | 1-shot 5-way |
|---|---|
| DOPE | 61.78 |
| DOPE w/o random background remove | 54.24 |
| DOPE w/o mask prediction | 50.06 |

## C    Additional Qualitative Results on CO3D

In Figure 1 we present additional qualitative results on the CO3D [20] dataset. We observe that our proposed approach can find correspondences between similar object parts across different instances of the same category.

36th Conference on Neural Information Processing Systems (NeurIPS 2022).

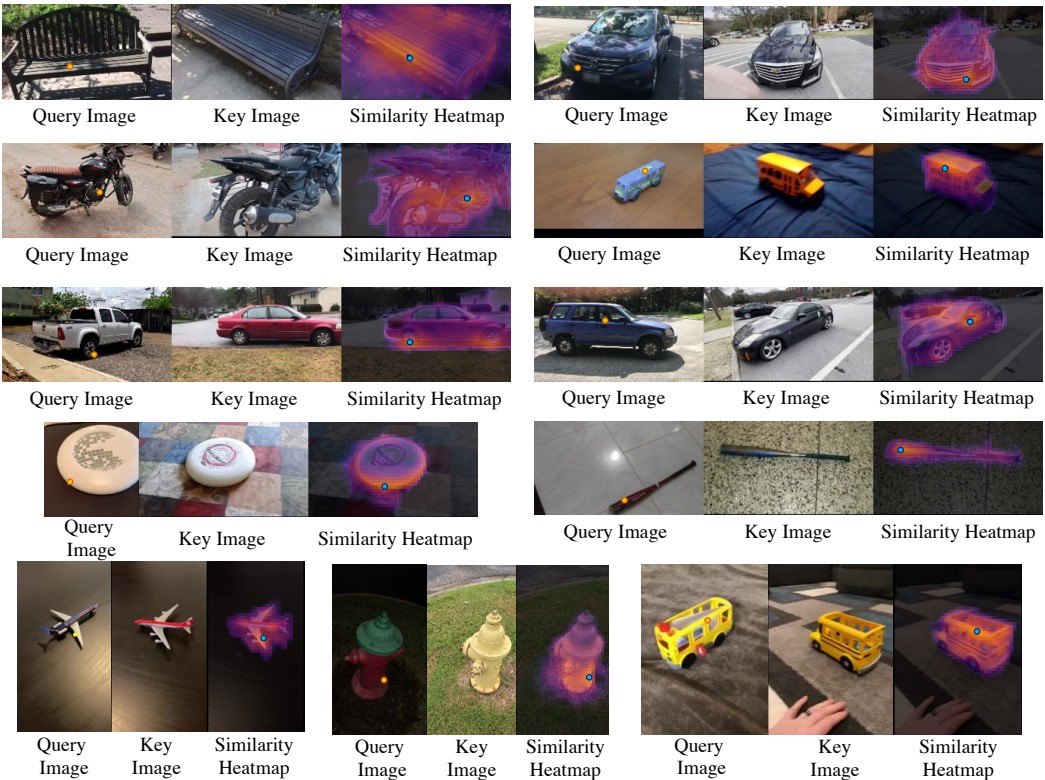

Figure 1: Additonal qualitative results on the CO3D dataset. We present a similarity heatmap for the query point (yellow) in the query image over the entire query image. The blue point on the heatmap overlaid image indicates the predicted correspondence of the query point. Note that these objects are from validation or test categories, which are different from the training categories.

# D Dataset Details

For all our synthetic datasets we render 20 views of each object randomly positioned on a plane with a physically based rendering (PBR) surface material that is randomly chosen. Lighting comes from a set of high dynamic range imaging (HDRI) lighting environments that are also randomly chosen. Rendering is done using the ray-tracing renderer Cycles in Blender [19]. We use 25 PBR materials and 46 HDRI maps with CC0 licenses sourced from PolyHaven [6]. We provide sample images from all synthetic datasets in Figure 2.

**Deriving Pixel-Level Correspondences** For the synthetic datasets ModelNet [26], ShapeNet [7], and ABC [14], we have ground truth camera instrinstics, extrinsics, depth maps and segmentation masks, which allows us to extract pixel-level correspondences between two views of an object. For the real CO3D dataset [20], each object video has camera instrinsics and extrinsics estimated using COLMAP [22, 21] and masks estimated with PointRend [13], which allows us to extract estimated pixel-level correspondences between two views of an object.

**Data Augmentation** For all self-supervised and low-shot learning algorithms, we perform color jittering, gamma, and contrast augmentations. In addition, we also randomly remove the background using the provided foreground mask. When the background is masked, we also randomly translate and rotate the foreground in the image, and randomize the background as in [10] (for examples please see Figure 3).

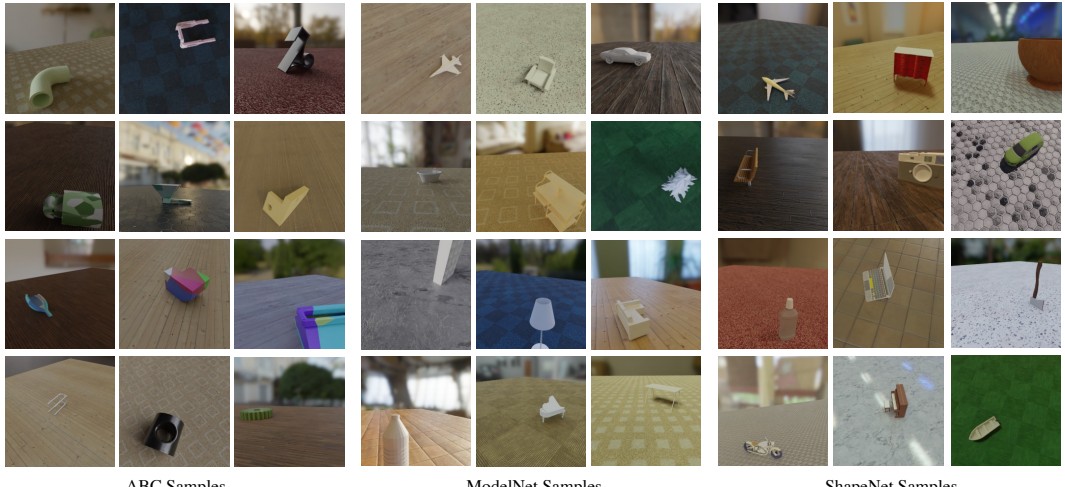

| ABC Samples | ModelNet Samples | ShapeNet Samples |

Figure 2: Visualization of our synthetic data rendered from ABC, ModelNet and ShapeNet. Note the high environment and viewpoint variability across the images.

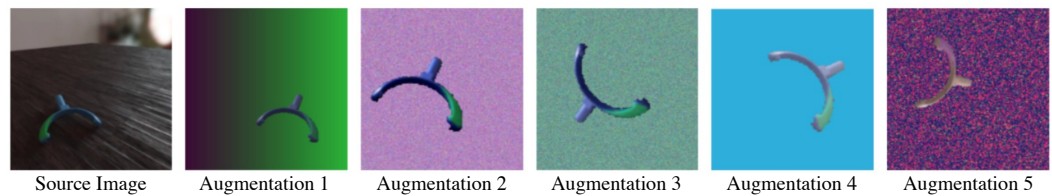

| Source Image | Augmentation 1 | Augmentation 2 | Augmentation 3 | Augmentation 4 | Augmentation 5 |

Figure 3: Illustration of our augmentation strategy by removing the background and applying random geometric transformations to the foreground object.

### D.1  ModelNet40-LS

The training and validation splits for ModelNet40-LS [26] are shown in Table 2, the first of which which we adopt from [23]. We use 15 queries for low-shot validation and testing. The dataset copyright information is available at `https://modelnet.cs.princeton.edu/`.

### D.2  ABC

We randomly sample 115K objects for training and 10K for validation from the total set of download-able objects. Originally the ABC [14] objects do not come with any surface materials. We generate materials with random colors and random Voronoi patterns when rendering the objects. The licensing information for this dataset is available at `https://deep-geometry.github.io/abc-dataset/#license`.

### D.3  ShapeNet-LS

The training and validation splits for ShapeNet55-LS [7] are shown in Table 3, the first of which we adopt from [23]. We use 15 queries for low-shot validation and testing. The dataset terms of use are available at `https://shapenet.org/terms`.

### D.4  CO3D-LS

The training and validation splits for CO3D-LS [20] are shown in Table 4. We select the training set by taking the 31 categories with the most data, and randomly sample two sets of 10 without replacement or validation and testing from the remaining 20 classes. For each object video clip in the dataset, we subsample every 3rd frame. We use 15 queries for low-shot validation and testing. The

terms and conditions of the CO3D dataset are available at `https://ai.facebook.com/datasets/co3d-downloads/`.

# E   Baseline Algorithm Implementation

All algorithms are implemented in PyTorch [17] 1.8.2 LTS where possible following released codebases from the original papers.

## E.1   SimpleShot

We follow the original implementation of SimpleShot [1]. We train SimpleShot [25] with the AdamW [15] optimizer, with a batch size of 256, a learning rate of 0.001, weight decay of 0.0001 as we found it gives improved results over using SGD with momentum. We train SimpleShot for 500 epochs on ShapeNet and 1000 epochs on CO3D and ModelNet with a 0.1 learning rate decay at 0.7 and 0.9 of the total number of epochs.

## E.2   RFS

We follow the original implementation of RFS from [2]. RFS [24] requires first training a backbone with cross-entropy on the training classes. To do this we follow the same training procedure as SimpleShot, as it also consists of training with cross-entropy on the base classes. Like in the original codebase, we use Scikit-Learn [18] to train a logistic classifier for each low-shot episode.

## E.3   FEAT

We follow the original implementation of FEAT [3, 27] in our implementation. We train a separate model for each $n$-way $m$-shot episode configuration as in the original paper. We use the original optimizer and hyperparameters, but halve the softmax temperature values for ShapeNet and ModelNet, and quarter them for CO3D as we find that it results in improved low-shot generalization.

## E.4   SupMoCo

We follow the pseudocode in the Appendix of the original paper to implement SupMoCo [16]. We use a queue of size $K = 4096$ because of our smaller datasets and SGD with cosine learning rate decay for 2000 epochs, with a batch size of 256, an initial learning rate of 0.05 and weight decay of 0.0001.

## E.5   VISPE

We use the original VISPE [12] implementation [4] in our experiments. We use AdamW [15] with a batch size of 32, learning rate of 0.0001, weight decay of 0.01 for 1000 epochs as we found it improves the low-shot generalization performance over the original hyperparameters.

## E.6   VISPE++

For our MoCo-based [11] VISPE++ baseline, we follow the original MoCo codebase [5]. Rather than the standard application of augmentations to a single view of an object to obtain two positive images, we give two views of the same object as two positive images, and two views of different objects as negatives. We use a two-layer 1024-dim MLP as the projection head. When training on ABC we use a queue of size $K = 16348$ and when training on other datasets we use a smaller queue of size $K = 4096$. We train this model using the AdamW [15] optimizer for 3500 epochs with a learning rate of 0.0001 and weight decay of 0.01.

For our SimSiam-based [8] VISPE++ baseline, we use the implementation from [9]. We use the same random masking augmentations as DOPE, but the original color jittering parameters of SimSiam because we found that results in improved low-shot generalization.

# F   Compute Details

To train our models we use an 8 GPU server with Titan Xp GPUs. Training our proposed approach requires 4 GPUs using Distributed Data Parallel in PyTorch [17].

| Training | # samples | Validation + Test | # samples | Split assignment |
|---|---|---|---|---|
| bookshelf | 672 | door | 129 | $v_0, t_1, t_2, t_3, t_4$ |
| chair | 989 | keyboard | 165 | $v_0, t_1, t_2, t_3, t_4$ |
| plant | 340 | flower_pot | 169 | $v_0, t_1, v_2, t_3, t_3$ |
| bed | 615 | curtain | 158 | $v_0, t_1, v_2, t_3, t_4$ |
| monitor | 565 | person | 108 | $v_0, v_1, t_2, v_3, t_4$ |
| piano | 331 | cone | 187 | $v_0, v_1, t_2, t_3, v_4$ |
| mantel | 384 | xbox | 123 | $v_0, v_1, v_2, t_3, t_4$ |
| car | 297 | cup | 99 | $v_0, t_1, t_2, v_3, t_4$ |
| table | 492 | bathtub | 156 | $v_0, v_1, v_2, v_3, t_4$ |
| bottle | 435 | wardrobe | 107 | $v_0, t_1, t_2, t_3, v_4$ |
| airplane | 726 | lamp | 144 | $t_0, v_1, t_2, v_3, v_4$ |
| sofa | 780 | stairs | 144 | $t_1, v_1, v_2, t_3, v_4$ |
| toilet | 444 | laptop | 169 | $t_0, t_1, t_2, v_3, t_4$ |
| vase | 575 | tent | 183 | $t_0, v_1, t_2, t_3, t_4$ |
| dresser | 286 | bench | 193 | $t_0, t_1, v_2, t_3, v_4$ |
| desk | 286 | range_hood | 215 | $t_0, t_1, t_2, v_3, v_4$ |
| night_stand | 286 | stool | 110 | $t_0, t_1, t_2, v_3, v_4$ |
| guitar | 255 | sink | 148 | $t_0, v_1, t_2, v_3, t_4$ |
| glass_box | 271 | radio | 124 | $t_0, v_1, v_2, v_3, t_4$ |
| tv_stand | 367 | bowl | 84 | $t_0, v_1, v_2, t_3, t_4$ |
| Total | | | | |
| 20 classes | 9396 | 20 classes | 2915 | |

Table 2: Split composition of ModelNet40-LS. Rightmost column indicates the assignment of the class to each of the 5 validation/testing splits.

| Training | # samples | Validation + Test | # samples | Split assignment |
|---|---|---|---|---|
| chair | 500 | stove | 218 | $v_0, v_1, t_2, t_3, v_4$ |
| table | 495 | microwave | 152 | $v_0, t_1, t_2, t_3, v_4$ |
| bathtub | 499 | microphone | 67 | $v_0, t_1, v_2, t_3, t_4$ |
| cabinet | 499 | cap | 56 | $v_0, v_1, t_2, v_3, v_4$ |
| lamp | 500 | dishwasher | 93 | $v_0, t_1, t_2, t_3, v_4$ |
| car | 525 | keyboard | 65 | $v_0, t_1, t_2, t_3, t_4$ |
| bus | 500 | tower | 133 | $v_0, v_1, t_2, t_3, t_4$ |
| cellular | 500 | helmet | 162 | $v_0, t_1, t_2, v_3, t_4$ |
| guitar | 500 | birdhouse | 73 | $v_0, t_1, v_2, t_3, v_4$ |
| bench | 499 | can | 108 | $v_0, t_1, t_2, t_3, t_4$ |
| bottle | 498 | piano | 239 | $t_0, v_1, t_2, v_3, t_4$ |
| laptop | 460 | train | 389 | $t_0, t_1, v_2, t_3, t_4$ |
| jar | 499 | file | 298 | $t_0, t_1, t_2, v_3, t_4$ |
| loudspeaker | 496 | pistol | 307 | $t_0, t_1, t_2, v_3, t_4$ |
| bookshelf | 452 | motorcycle | 337 | $t_0, t_1, v_2, t_3, t_4$ |
| faucet | 500 | printer | 166 | $t_0, t_1, t_2, t_3, v_4$ |
| vessel | 864 | mug | 214 | $t_0, v_1, t_2, t_3, t_4$ |
| clock | 496 | rocket | 85 | $t_0, v_1, v_2, t_3, t_4$ |
| airplane | 500 | skateboard | 152 | $t_0, v_1, v_2, v_3, v_4$ |
| pot | 500 | bed | 233 | $t_0, t_1, t_2, t_3, v_4$ |
| rifle | 498 | ashcan | 343 | $t_0, t_1, t_2, t_3, v_4$ |
| display | 498 | washer | 169 | $t_0, t_1, t_2, t_3, t_4$ |
| knife | 423 | bowl | 186 | $t_0, t_1, v_2, t_3, t_4$ |
| telephone | 498 | bag | 83 | $t_0, v_1, v_2, v_3, t_4$ |
| sofa | 499 | mailbox | 94 | $t_0, v_1, t_2, t_3, t_4$ |
| | | pillow | 96 | $t_0, t_1, t_2, t_3, t_4$ |
| | | earphone | 73 | $t_0, t_1, v_2, t_3, t_4$ |
| | | camera | 113 | $t_0, t_1, t_2, t_3, t_4$ |
| | | basket | 113 | $t_0, v_1, v_2, v_3, v_4$ |
| | | remote | 66 | $t_0, t_1, t_2, t_3, t_4$ |
| Total | | | | |
| 25 classes | 12698 | 30 classes | 4883 | |

Table 3: Split composition of ShapeNet55-LS. Rightmost column indicates the assignment of the class to each of the 5 validation/testing splits.

| Training | # samples | Validation + Test | # samples | Split assignment |
|---|---|---|---|---|
| wineglass | 453 | car | 210 | $v_0, t_1, t_2, t_3, v_4$ |
| keyboard | 638 | bottle | 277 | $v_0, v_1, v_2, t_3, t_4$ |
| mouse | 431 | baseballglove | 84 | $v_0, v_1, t_2, t_3, v_4$ |
| bowl | 660 | frisbee | 121 | $v_0, t_1, t_2, t_3, t_4$ |
| broccoli | 379 | tv | 29 | $v_0, v_1, v_2, v_3, v_4$ |
| chair | 675 | toyplane | 225 | $v_0, v_1, v_2, t_3, t_4$ |
| handbag | 749 | baseballbat | 71 | $v_0, t_1, t_2, v_3, t_4$ |
| toytrain | 272 | pizza | 134 | $v_0, v_1, t_2, v_3, v_4$ |
| carrot | 740 | hydrant | 307 | $v_0, t_1, t_2, v_3, v_4$ |
| bicycle | 340 | hotdog | 69 | $v_0, v_1, v_2, v_3, t_4$ |
| cellphone | 416 | parkingmeter | 21 | $t_0, t_1, v_2, t_3, t_4$ |
| ball | 542 | banana | 198 | $t_0, v_1, v_2, t_3, v_4$ |
| teddybear | 734 | motorcycle | 267 | $t_0, t_1, t_2, t_3, v_4$ |
| cake | 348 | bench | 250 | $t_0, v_1, t_2, t_3, v_4$ |
| backpack | 832 | donut | 193 | $t_0, t_1, v_2, v_3, t_4$ |
| hairdryer | 503 | microwave | 50 | $t_0, v_1, t_2, v_3, t_4$ |
| couch | 223 | stopsign | 193 | $t_0, t_1, v_2, t_3, v_4$ |
| toilet | 355 | skateboard | 82 | $t_0, t_1, v_2, v_3, t_4$ |
| remote | 392 | toybus | 141 | $t_0, v_1, t_2, v_3, t_4$ |
| toaster | 299 | kite | 150 | $t_0, t_1, v_2, v_3, v_4$ |
| vase | 647 | | | |
| laptop | 501 | | | |
| toytruck | 466 | | | |
| umbrella | 498 | | | |
| suitcase | 482 | | | |
| plant | 563 | | | |
| apple | 391 | | | |
| cup | 169 | | | |
| book | 658 | | | |
| sandwich | 244 | | | |
| orange | 479 | | | |
| Total | | | | |
| 31 classes | 15079 | 20 classes | 3072 | |

Table 4: Split composition of CO3D. Rightmost column indicates the assignment of the class to each of the 5 validation/testing splits.

# Appendix References

[1] `https://github.com/mileyan/simple_shot`.

[2] `https://github.com/WangYueFt/rfs/`.

[3] `https://github.com/Sha-Lab/FEAT`.

[4] `https://github.com/chihhuiho/VISPE`.

[5] `https://github.com/facebookresearch/moco`.

[6] Poly haven, https://polyhaven.com/.

[7] Angel X Chang, Thomas Funkhouser, Leonidas Guibas, Pat Hanrahan, Qixing Huang, Zimo Li, Silvio Savarese, Manolis Savva, Shuran Song, Hao Su, et al. Shapenet: An information-rich 3d model repository. *arXiv preprint arXiv:1512.03012*, 2015.

[8] Xinlei Chen and Kaiming He. Exploring simple siamese representation learning. In *Proceedings of the IEEE/CVF Conference on Computer Vision and Pattern Recognition*, pages 15750–15758, 2021.

[9] MMSelfSup Contributors. MMSelfSup: Openmmlab self-supervised learning toolbox and benchmark. `https://github.com/open-mmlab/mmselfsup`, 2021.

[10] Peter R Florence, Lucas Manuelli, and Russ Tedrake. Dense object nets: Learning dense visual object descriptors by and for robotic manipulation. In *Conference on Robot Learning*, pages 373–385. PMLR, 2018.

[11] Kaiming He, Haoqi Fan, Yuxin Wu, Saining Xie, and Ross Girshick. Momentum contrast for unsupervised visual representation learning. In *Proceedings of the IEEE/CVF conference on computer vision and pattern recognition*, pages 9729–9738, 2020.

[12] Chih-Hui Ho, Bo Liu, Tz-Ying Wu, and Nuno Vasconcelos. Exploit clues from views: Self-supervised and regularized learning for multiview object recognition. In *Proceedings of the IEEE/CVF Conference on Computer Vision and Pattern Recognition*, pages 9090–9100, 2020.

[13] Alexander Kirillov, Yuxin Wu, Kaiming He, and Ross Girshick. Pointrend: Image segmentation as rendering. In *Proceedings of the IEEE/CVF conference on computer vision and pattern recognition*, pages 9799–9808, 2020.

[14] Sebastian Koch, Albert Matveev, Zhongshi Jiang, Francis Williams, Alexey Artemov, Evgeny Burnaev, Marc Alexa, Denis Zorin, and Daniele Panozzo. Abc: A big cad model dataset for geometric deep learning. In *Proceedings of the IEEE Conference on Computer Vision and Pattern Recognition*, pages 9601–9611, 2019.

[15] Ilya Loshchilov and Frank Hutter. Decoupled weight decay regularization. In *International Conference on Learning Representations*, 2018.

[16] Orchid Majumder, Avinash Ravichandran, Subhransu Maji, Alessandro Achille, Marzia Polito, and Stefano Soatto. Supervised momentum contrastive learning for few-shot classification. *arXiv preprint arXiv:2101.11058*, 2021.

[17] Adam Paszke, Sam Gross, Francisco Massa, Adam Lerer, James Bradbury, Gregory Chanan, Trevor Killeen, Zeming Lin, Natalia Gimelshein, Luca Antiga, et al. Pytorch: An imperative style, high-performance deep learning library. *Advances in neural information processing systems*, 32, 2019.

[18] F. Pedregosa, G. Varoquaux, A. Gramfort, V. Michel, B. Thirion, O. Grisel, M. Blondel, P. Prettenhofer, R. Weiss, V. Dubourg, J. Vanderplas, A. Passos, D. Cournapeau, M. Brucher, M. Perrot, and E. Duchesnay. scikit-learn: Machine learning in Python. *Journal of Machine Learning Research*, 12:2825–2830.

[19] Blender Project. `https://blender.org`.

[20] Jeremy Reizenstein, Roman Shapovalov, Philipp Henzler, Luca Sbordone, Patrick Labatut, and David Novotny. Common objects in 3d: Large-scale learning and evaluation of real-life 3d category reconstruction. In *Proceedings of the IEEE/CVF International Conference on Computer Vision*, pages 10901–10911, 2021.

[21] Johannes Lutz Schönberger and Jan-Michael Frahm. Structure-from-motion revisited. In *Conference on Computer Vision and Pattern Recognition (CVPR)*, 2016.

[22] Johannes Lutz Schönberger, Enliang Zheng, Marc Pollefeys, and Jan-Michael Frahm. Pixelwise view selection for unstructured multi-view stereo. In *European Conference on Computer Vision (ECCV)*, 2016.

[23] Stefan Stojanov, Anh Thai, and James M. Rehg. Using shape to categorize: Low-shot learning with an explicit shape bias. In *Proceedings of the IEEE/CVF Conference on Computer Vision and Pattern Recognition (CVPR)*, pages 1798–1808, June 2021.

[24] Yonglong Tian, Yue Wang, Dilip Krishnan, Joshua B Tenenbaum, and Phillip Isola. Rethinking few-shot image classification: a good embedding is all you need? In *European Conference on Computer Vision (ECCV) 2020*, August 2020.

[25] Yan Wang, Wei-Lun Chao, Kilian Q Weinberger, and Laurens van der Maaten. Simpleshot: Revisiting nearest-neighbor classification for few-shot learning. *arXiv preprint arXiv:1911.04623*, 2019.

[26] Zhirong Wu, Shuran Song, Aditya Khosla, Fisher Yu, Linguang Zhang, Xiaoou Tang, and Jianxiong Xiao. 3d shapenets: A deep representation for volumetric shapes. In *Proceedings of the IEEE conference on computer vision and pattern recognition*, pages 1912–1920, 2015.

[27] Han-Jia Ye, Hexiang Hu, De-Chuan Zhan, and Fei Sha. Few-shot learning via embedding adaptation with set-to-set functions. In *Proceedings of the IEEE/CVF Conference on Computer Vision and Pattern Recognition (CVPR)*, June 2020.