# OpenReview forum: "Learning Dense Object Descriptors from Multiple Views for Low-shot Category Generalization"
_NeurIPS.cc/2022/Conference — NeurIPS 2022 Accept_

### Official Review · Reviewer_tG2S · 2022-06-24

**Rating:** 7
**Confidence:** 2
**Soundness:** 3 good
**Presentation:** 4 excellent
**Contribution:** 2 fair

**Summary:**

-- Post rebuttal update --
The authors have addressed all the questions and weaknesses in my original review using new experiments. In particular the main thesis of the paper has become clear to me now, that it is a comparison between geometric and semantic supervision for low shot learning. The authors have agreed to change their title and description to reflect that the method is patch level and not part level. New experiments show that the methods is much better than Vader and the message remains the same even after supervised baselines are given the same data augmentation. Multi object experiments show that the method is not severely limited to single object datasets as I had originally anticipated. I have updated my rating to 7.
-- End of post rebuttal update ---

This paper presents a method for learning grid structured embeddings from images, called DOPE.

DOPE leverages pairs of images which are two views of a single object with known foreground segmentation, depth maps, camera parameters and camera poses. This information is used to derive pixel level correspondence between the two images which create a positive pair for contrastive learning. Other pairs are treated as negatives. Contrastive learning is used to learn the required embeddings which exhibit the following properties.

1. Embeddings can be used to match across instances of the same category.
2. Such properties extend to novel classes that were not used in training.

These properties combined are used to do nearest neighbor classification to demonstrate SoTA low-shot classification.

**Questions:**

### More questions
- Why use the predicted foreground mask, \hat{M}, instead of the ground truth mask, M, in line 182. Same in line 192?

- Is PointRend in line 236 unsupervised / self-supervised?

### Minor comments
- Please discuss the observation that local embeddings work better than global ones again in the experiments. Which two rows should the reader compare to know the performance gap? Is this an apples-to-apples comparison?

- In section 2.1, please state which bucket DOPE falls into. "Metric-learning".

- Do supervised baselines also get the random background masking data augmentation?

### Typos
- Line 2: datasets for training feature representations
- Line 151: For illustration 'p'lease refer to Figure 2.
- Line 245: trained on the support data

**Limitations:**

Limitations around data simplicity need to be discussed in section 7. Can DOPE be pre-trained using say the epic kitchens dataset?

Societal impact has been adequately discussed focusing on the environmental effects of training these models.

**Strengths And Weaknesses:**

### Strengths

- The main thesis of this paper is that local features matching provides better downstream generalization for low-shot recognition. This has been supported by comparing against baselines.

- Results include both real and synthetic data and compare against both supervised and self-supervised prior work.

- Qualitative visualizations show that the embedding is indeed local. For instance the front of the motorcycle in figure 5 has a different embedding than the middle and back of the motorcycle. Such embeddings might be useful for more fine-grained tasks in future work.

- The tables show that DOPE performs well, beating even supervised baselines on the ModelNet40-LS dataset.

### Weaknesses and questions.
- While the method does not use category labels for self-supervised pre-training, it requires foreground segmentation, depth maps, camera parameters and camera pose. This is very rich information which they derive using COLMAP for the CO3D-LS dataset, but that might not be work well for in-the-wild large scale video corpora. This limitation has been discussed in section 7.

- The paper refers to embeddings as capturing object parts. However, nothing in the formulation is part specific, rather it goes down to the pixel or patch level. The visualization show a continuous variation from one point on the object to another as opposed to a part level segmentation emerging from the algorithm. The presentation would be much clearer if the paper referred to these as patch embeddings from the beginning. Both the paper title and the method name convey that it is "part" based which is confusing.

- All experiments and the formulation assume that there is only one object in the scene. This wasn't discussed explicitly even in section 7. Specifically, what can we expect from DOPE when trained in cluttered natural environments?

- Missing comparison against VADeR (SimCLR style contrastive learning but at the pixel level using two views generated by data augmentation) [1], FLow-E (Similar to VADeR but using BYOL style loss instead of SimCLR) [2], DenseCL [3] or PixPro [4], some recent local embedding learning methods. In fact the architecture borrows from VADeR. FLow-E code is not available. But PixPro has public code [here](https://github.com/zdaxie/PixPro), DenseCL [here](https://github.com/WXinlong/DenseCL). Basically what does a SoTA self-supervised local-embedding learning method do on these datasets? On a related note, how does DOPE perform when the two views come from data augmentation?

[1] Pedro O O Pinheiro, Amjad Almahairi, Ryan Benmalek, Florian Golemo, and Aaron C Courville. Unsupervised learning of dense visual representations. NeurIPS 2020.

[2] Xiong, Yuwen, et al. "Self-supervised representation learning from flow equivariance." Proceedings of the IEEE/CVF International Conference on Computer Vision. 2021.

[3] Wang, Xinlong, et al. "Dense contrastive learning for self-supervised visual pre-training." Proceedings of the IEEE/CVF Conference on Computer Vision and Pattern Recognition. 2021.

[4] Xie, Zhenda, et al. "Propagate yourself: Exploring pixel-level consistency for unsupervised visual representation learning." Proceedings of the IEEE/CVF Conference on Computer Vision and Pattern Recognition. 2021.

---

> ### Author Response · Authors · 2022-08-02
> **To Reviewer tG2S - Minor Comments Answers**
>
> ### Minor Comments
>
> **R4 Q8:** *Please discuss the observation that local embeddings work better than global ones again in the experiments. Which two rows should the reader compare to know the performance gap? Is this an apples-to-apples comparison?*
>
> **R4 A8:**
> *Local vs Global Comparison*: This is a fair comparison, but it is not an apples-to-apples comparison.
> -   DOPE needs estimated sparse depth and calibrated cameras for training, whereas VISPE++ does not. However, these can be sufficiently estimated using geometric optimization pipelines like COLMAP, which we successfully use in our experiments on real data. Prior methods cannot take advantage of this additional supervision, and our work aims to understand whether and how this additional signal can be utilized and show its benefits for few-shot category generalization
> -   Both DOPE and VISPE++ require known masks for training. Without known masks, random background removal cannot be done, and both models learn to trivially tell instances apart because of the backgrounds and not the objects themselves.
>
> *Row comparisons* We present these results here again, originally found in the last two rows of Table 1 and Table 2 in the main text and Table 2 of the supplement.
>
> CO3D-LS - Real
> | | 1-shot 5-way | 5-shot 5-way| 1-shot 10-way | 5-shot 10-way |
> | --| ----------- | ----------- | ------------- | ----------------|
> VISPE++  | 56.82 (0.40) | 72.61 (0.31) | 44.02 (0.25) | 61.22 (0.17) |
> DOPE  | 58.31 (0.40) | 73.40 (0.15) | 44.91 (0.15) | 61.40 (0.08) |
>
> Modelnet40-LS - Synthetic
> | | 1-shot 5-way | 5-shot 5-way| 1-shot 10-way | 5-shot 10-way |
> | --| ----------- | ----------- | ------------- | ----------------|
> VISPE++ -- ABC | 56.50 (0.43) | 69.75 (0.32) | 42.97 (0.24) | 61.61 (0.16) |
> DOPE -- ABC | 57.50 (0.42) | 71.56 (0.31) | 44.14 (0.23) | 59.14 (0.23) |
>
> ShapeNet55-LS - Synthetic
> | | 1-shot 5-way | 5-shot 5-way| 1-shot 10-way | 5-shot 10-way |
> | --| ----------- | ----------- | ------------- | ----------------|
> VISPE++ -- ABC | 56.86 (0.44) | 69.08 (0.35) | 42.53 (0.24) | 55.13 (0.24) |
> DOPE - ABC | 58.31 (0.40)  | 73.40 (0.15) | 44.91 (0.15)  | 61.40 (0.08) |
>
> We hypothesize that global encodings are not necessarily useful for low-shot category generalization, because to determine whether two object instances are similar or not, global encodings only need to contain minimal information e.g. only a few object properties. In contrast, learning at the local level about the similarity of object patches forces the model to appropriately encode all visible object properties in a pair of views. This is more useful for low-shot category generalization because a local model has learned a richer set of features.
>
> ---
>
> **R4 Q9:** *In section 2.1, please state which bucket DOPE falls into. "Metric-learning".*
>
> **R4 A9:** This is correct, DOPE falls into the Metric Learning body of related work. We will update the draft to reflect this.
>
> ---
> **R4 Q10:** *Do supervised baselines also get the random background masking data augmentation?*
>
> **R4 A10:** This is a great question. In fact, supervised baselines did not get the random background masking data augmentation. We remedied this issue with additional experiments. We present results below for SimpleShot and RFS trained with random background masking on ModelNet below, and find that it generally results in a <1% performance improvement. While the findings of our paper still remain the same in light of this small improvement, for the final version we will re-train the baselines using this augmentation strategy for a more direct comparison.
> | | 1-shot 5-way | 5-shot 5-way| 1-shot 10-way | 5-shot 10-way |
> | --| ----------- | ----------- | ------------- | ----------------|
> SimpleShot -- Without Masking | 50.84 (0.41) | 63.67 (0.33)  | 36.45 (0.22) |49.89 (0.16)  |
> SimpleShot -- With Masking | 51.25 (0.29) | 64.28 (0.23) | 37.70 (0.16) | 50.75 (0.11) |
> RFS -- Without Masking | 50.85 (0.40) | 66.48 (0.35) | 37.11 (0.16) | 53.37 (0.14) |
> RFS -- With Masking | 51.42 (0.41) |  67.34 (0.34) | 37.92 (0.23) | 54.69 (0.16) |
>
> ---

---

> ### Author Response · Authors · 2022-08-02
> **To Reviewer tG2S - Answers to More Questions**
>
> ### More Questions
>
> **R4 Q6:**  Why use the predicted foreground mask, \hat{M}, instead of the ground truth mask, M, in line 182. Same in line 192?
> **R4 A6:** We do not assume that ground truth masks are available for images used at inference time e.g. during low-shot classification. Therefore, when masking the predicted feature grid, we use the predicted foreground mask.
>
> ---
>
> **R4 Q7:** *Is PointRend in line 236 unsupervised / self-supervised? (can be separated on its own)*
>
> **R4 A7:** PointRend is trained with full supervision on COCO.

---

> ### Author Response · Authors · 2022-08-02
> **To Reviewer tG2S - Answers to Weaknesses Continued**
>
> ### Weaknesses Continued
>
> **R4 Q3:** *All experiments and the formulation assume that there is only one object in the scene. This wasn't discussed explicitly even in section 7. Specifically, what can we expect from DOPE when trained in cluttered natural environments?*
>
> **R4 A3:**
>
> 1) Our experiments feature images with a single foreground object, but we note that this is true of the majority of prior research on large scale and low-shot object classification, including the widely-used ImageNet and CIFAR datasets. We feel this setting is an important first step in establishing the feasibility of our approach.
>
> 2) Motivated by this question, we conducted an experiment to investigate the performance of our method at finding similarity between objects when images contain multiple objects. The qualitative examples [shown at the following link](https://imgur.com/a/Flbjim7) demonstrate that DOPE generalizes well to scenes that consist of multiple objects, either from different viewpoints or different scene settings (varied background, lighting, object textures, scales and poses). For instance, DOPE was able to identify the correspondences in the first example in the “Same Objects, Different Scene” set of visualizations, where the object’s pose and texture change significantly and is partly occluded by another object.
>
> ---
>
> **R4 Q5:** *Missing comparison against VADeR (SimCLR style contrastive learning but at the pixel level using two views generated by data augmentation) [1], FLow-E (Similar to VADeR but using BYOL style loss instead of SimCLR) [2], DenseCL [3] or PixPro [4], some recent local embedding learning methods. In fact the architecture borrows from VADeR. FLow-E code is not available. But PixPro has public code [here](https://github.com/zdaxie/PixPro), DenseCL [here](https://github.com/WXinlong/DenseCL). Basically what does a SoTA self-supervised local-embedding learning method do on these datasets? On a related note, how does DOPE perform when the two views come from data augmentation?*
>
> **R4 A5:** The reviewer suggested a comparison to purely 2D dense contrastive learning methods like VaDeR, PixPro and DenseCL when they are trained on two views that come from data augmentations of the same image. We carefully investigated this suggestion:
>
> *VaDeR*: We conducted an additional comparison to VaDeR, using an improved negative sampling strategy that we found to lead to better performance (where negatives come from the same object rather than different objects and without using a queue of negatives) and added a mask prediction as used in DOPE. These modifications allow for a more fair comparison. The data augmentations and low-shot prediction procedure used by both methods are identical. We observe that the lack of multi-view information results in a significant performance gap for VaDeR.
>
> ModelNet40-LS Validation Set
> | | 1-shot 5-way |
> | --| ----------- |
> DOPE - trained with multi view images | 69.37 |
> VaDeR - trained with 2D augmented images | 43.90 |
>
> *DenseCL:* We attempted to train DenseCL on ABC, and evaluate its low-shot generalization performance on the ModelNet40-LS validation set. As DenseCL does not perform mask prediction, we gave it the advantage of always having ground truth masks, which allowed us to use the same low-shot prediction procedure as dope. We found that low-shot generalization performance was poor at 22% accuracy. We hypothesize that this is due to the lack of FPN in DenseCL, resulting in a spatially very coarse 7x7 representation as opposed to the FPNs 56x56, which doesn’t allow for all object properties visible in the image to be encoded properly.
>
> *PixPro:* This work has a contrastive pre-trained FPN in the paper, but code for pre-training using an FPN is not included in the code release provided by the authors. We attempted training PixPro on ABC with a standard ResNet but could not get the model to converge. Further, [there have been issues](https://github.com/zdaxie/PixPro/issues/14) raised regarding the reproducibility of the results using this codebase.
>
> *In Summary:* We performed a study to compare with 2D dense contrastive learning. All of these methods (the more recent as well as the older ones) are within a few mAP points in performance on downstream tasks. The observed large gap between multi-view and 2D indicates that the algorithm design improvements of the more recent methods cannot make up for the lack of multi-view information. We will incorporate the VaDeR results in the paper to quantify the additional benefit of multi-view self-supervision.
>
> ---

---

> > ### Comment · Reviewer_tG2S · 2022-08-08
> > **Reply**
> >
> > Multi object results: Please include these in the paper / supplementary.
> >
> > VaDeR comparison: Thank you for running these experiments. Indeed multi-view supervision is much stronger than anticipated. What are the results when the sampling is not changed?

---

> > > ### Author Response · Authors · 2022-08-09
> > > **Reply to tG2S**
> > >
> > > Following the reviewer's suggestion, we will include the multi object results in the final version.
> > >
> > > We previously experimented with both sampling strategies, and having found the one with better performance, in the interest of time we focused on using it for the results we presented in the rebuttal. In the final version of the paper we will include a direct comparison to quantify the differences.
> > >
> > > Please let us know if there is any more information we can provide that might impact your final rating.

---

> > > > ### Comment · Reviewer_tG2S · 2022-08-09
> > > > **Reply**
> > > >
> > > > I think this sufficiently addresses my concerns. The paper is considerably improved post rebuttal. I will update my rating.

---

> ### Author Response · Authors · 2022-08-02
> **To Reviewer tG2S - Answers to Weaknesses Continued**
>
> **R4 Q2:** *The paper refers to embeddings as capturing object parts. However, nothing in the formulation is part specific, rather it goes down to the pixel or patch level. The visualization shows a continuous variation from one point on the object to another as opposed to a part level segmentation emerging from the algorithm. The presentation would be much clearer if the paper referred to these as patch embeddings from the beginning. Both the paper title and the method name convey that it is "part" based which is confusing..*
>
> **R4 A2:**  We thank the reviewer for this suggestion and agree that the clarity will be improved if we refer to the embeddings learned by our model as “patch embeddings”. We will adjust the writing throughout the draft to reflect this, rename the method to “DOPE: Deep Object Patch Embedding”, and change the paper’s title to “Learning Object Patches from Multiple Views for Low-shot Category Generalization”

---

> > ### Comment · Reviewer_tG2S · 2022-08-08
> > **Reply**
> >
> > Thank you for incorporating this feedback.

---

> ### Author Response · Authors · 2022-08-02
> **To Reviewer tG2S - Answers to Weaknesses**
>
> ### Weaknesses
> **R4 Q1:** *While the method does not use category labels for self-supervised pre-training, it requires foreground segmentation, depth maps, camera parameters and camera pose. This is very rich information which they derive using COLMAP for the CO3D-LS dataset, but that might not work well for in-the-wild large scale video corpora.*
>
> **R4 A1:**
>
> 1) A basic question is whether a self-supervised category learner can be trained without any category labels during training time. We are the first to definitively establish that this can be done, using multi-view self-supervision, and moreover our performance beats state of the art that require category labels during training on synthetic data, and is competitive on real data.
>
> 2) The reviewer asks about the source of all the geometric inputs that we require. The sparse depth maps, camera parameters and camera pose used by our method can be easily obtained without human labeling--- from the data renderer in the synthetic case and off-the-shelf geometric methods (COLMAP) in the case of real-world data (CO3D-LS). There is a question however of what is needed for object segmentation. Masks are directly obtained from the renderer in the synthetic case, and by running a COCO pretrained PointRend on CO3D-LS (real images). Inspired by the reviewer’s question, we conducted an additional experiment that shows that our method can still succeed when the segmentation masks themselves are obtained using an off the shelf unsupervised instance segmentation method, freeSOLO [A]. (We did this for ABC under the time pressure of the rebuttal by directly using the pre-trained freeSOLO, and will add results on CO3D-LS in the final version). This step completely removes any indirect dependence on image labels from our solution approach.
>      - We present results for training for DOPE and VISPE++ on ABC with ground truth (GT) masks and freeSOLO predicted masks (see below). We test low-shot generalization on ModelNet. The category supervised RFS is provided as reference. We observe that while performance decreases for DOPE when it is trained with freeSOLO masks as opposed to GT masks, the performance is still competitive with category supervised methods. Note that there is a domain gap between our data and the freeSOLO training data and that training freeSOLO on our data domains for mask estimation in the final version should lead to improved performance.
>
> | | 1-shot 5-way | 5-shot 5-way| 1-shot 10-way | 5-shot 10-way |
> | --| ----------- | ----------- | ------------- | ----------------|
> RFS - Category Supervised| 50.85 (0.40) | 66.48 (0.35) | 37.11 (0.33) | 53.37 (0.16) |
> VISPE++ -- ABC with GT mask |  58.31 (0.40) | 73.40 (0.15) | 44.91 (0.15) | 61.40 (0.08) |
> VISPE++ -- ABC with freeSOLO mask | 50.28 (0.31) | 64.95 (0.33) | 36.80 (0.22) | 51.37 (0.18) |
> DOPE -- ABC with GT mask | 56.82 (0.40) | 72.61 (0.31) | 44.02 (0.25) | 61.22 (0.17) |
> DOPE -- ABC with freeSOLO mask | 52.34 (0.40) | 66.17 (0.32) | 38.87 (0.22) | 52.78 (0.18) |
>
>
> 3) The reviewer asks whether our approach could work “in-the-wild” based on a concern about whether CO3D-LS is a challenging real-world dataset. First, at present CO3D-LS is the only available real world dataset for evaluating multi-view self-supervised learning for low-shot object categorization. Second, we want to emphasize that CO3D-LS is challenging: It consists of crowdsourced videos taken with mobile devices, in both indoor and outdoor settings, that often have motion blur and contain multiple objects in addition to the object of interest. We want to emphasize that the sparse depth maps, camera parameters and camera pose are generated using the off shelf geometric optimization method COLMAP, which produces estimates significantly noisier than in the synthetic setting, which DOPE can successfully handle. The visualizations of CO3D [at this link](https://imgur.com/a/iFyzlEy) demonstrate the challenges of this dataset.
>
> [A] Wang, Xinlong, et al. "FreeSOLO: Learning to Segment Objects without Annotations." Proceedings of the IEEE/CVF Conference on Computer Vision and Pattern Recognition. 2022.

---

> > ### Comment · Reviewer_tG2S · 2022-08-08
> > **Reply**
> >
> > Thank you for sharing the visualizations of CO3D and the experiments using freeSOLO. While the dataset is not as complex as random youtube videos, it appears to be sufficiently complex to support the thesis of this paper. In the context of additional clarification provided in the general response above about contributions and novelty, demonstrating in-the-wild learning capabilities is not as relevant anymore.

---

### Official Review · Reviewer_BZQB · 2022-07-12

**Rating:** 5
**Confidence:** 4
**Soundness:** 3 good
**Presentation:** 3 good
**Contribution:** 3 good

**Summary:**

This paper proposes a novel method for low-shot category generalization via the object part correspondence learning. The object part correspondence is learnt by contrastive learning on multiple views of object instances. The anchors are constructed by positive features at pixel locations of the same 3D point on the object surface and negatives features from different points on the object surface. To obtain the dense object feature, the autors propose a Deep Object Part Encoding (DOPE), which is a simple framework with the Panoptic FPN. During the inference, several keypoints of the shot image are first sampled and extract the pixel-level feature by DOPE. Then, the keypoint features are compared with all pixel feature in gallery image to obtain the similarity score. Finally, the nearest neighbor method is used to determine the category.

**Questions:**

As mentioned before, the proposed method requires more information, i.e., depth and camera pose. Although these information are free for synthetic data, it still not very easy to capture for real images. I wonder if you have tried to transfer the model trained on synthetic data to the real images directly or with some tricks. I think these results could be interesting and if the generalization ability is not so bad, the contribution of this work would be improved.

**Strengths And Weaknesses:**

Strengths:

The proposed method is well-defined and esay to understand and it outperforms the state-of-the-arts approachs on few shot classification.

The authors conduct comprehensive experiments on several benchmarks including both synthetic data and real-world data. And the consistent improvement can be observed on different few shot learning setting.

Weakness:

In Figure4, the k pixels on the object surface are randomly sampled from each shot image and compare them with all pixel-wise feature of query image, however, from Line #192 to Line #200, the k pixels are sampled from the query image while compare them with all pixel-wise feature of  the shot image. These two description are conflict, and whick one is correct?

Although this paper claims it doesn't require any category-level annotations, but it still need lots of additional supervision signal, i.e., foreground mask, depth map and camera pose during training. I think it is unfair to compare the proposed method with the existing work. Especially, the depth map actually involves lots of geometry information than the pure RGB image.

---

> ### Author Response · Authors · 2022-08-02
> **To Reviewer BZQB - Answers to Weaknesses and Questions**
>
> ### Weaknesses
>
> **R3 Q1:** *In Figure4, the k pixels on the object surface are randomly sampled from each shot image and compare them with all pixel-wise feature of query image, however, from Line #192 to Line #200, the k pixels are sampled from the query image while compare them with all pixel-wise feature of the shot image. These two description are conflict, and whick one is correct?*
>
> **R3 A1:** Thank you for pointing this out. We apologize for the inconsistency. The correct description is the one from L192-L200. We will make the appropriate fixes for this in the final version.
>
> ---
>
> **R3 Q2:** *Although this paper claims it doesn't require any category-level annotations, but it still need lots of additional supervision signal, i.e., foreground mask, depth map and camera pose during training. I think it is unfair to compare the proposed method with the existing work. Especially, the depth map actually involves lots of geometry information than the pure RGB image.*
>
> **R3 A2:**
> 1) For a discussion of category vs geometric supervision please see our response to **R4 Q1**
> 2) Fairness: We note that our method and the competing methods all start with just RGB images as input. We believe it is fair to automatically extract geometric information as part of an approach based on multi-view self-supervision. This comparison demonstrates the ability of geometric self-supervision to compete effectively with category labels in training few shot models.
> ---
> ### Questions
> **R3 Q3**: *As mentioned before, the proposed method requires more information, i.e., depth and camera pose. Although these information are free for synthetic data, it still not very easy to capture for real images*
>
> **R3 A3**:
> 1) Note that our method does not use depth maps as input, but it requires them to derive correspondences during training
> 2) Our experiments with CO3D-LS demonstrate that our model can train with noisy and sparse estimated depth maps and camera pose on real data, automatically estimated using COLMAP (please refer to [this link](https://imgur.com/a/iFyzlEy) for a visualization of the estimated masks and depth maps of CO3D).

---

### Official Review · Reviewer_3MZa · 2022-07-12

**Rating:** 6
**Confidence:** 4
**Soundness:** 3 good
**Presentation:** 3 good
**Contribution:** 3 good

**Summary:**

The paper proposes a self-supervised learning task to perform similarity metric learning across multiple view for the local representation of an object with the assistance of sparse depths, foreground masks, and known camera parameters (from COLMAP.) It achieves superior performance even better than some supervised baseline methods for low-shot classification of novel categories.

**Questions:**

It is great that the authors show that to enforcing the self-supervised loss for the local invariant representation of an object across different view can benefit low-shot classification. However, I have few questions:

1. How are the views sampled for the training? If the object views are too extreme, the estimation of COLMAP may fail.
2. How do the preprocessing errors from the camera parameters, sparse depth, and segmentation map affect the performance? Although w in appendix, the authors do provide the ablation studies of w/o random background remove and w/o mask prediction.
3. Is the 3D-based self-supervision necessary since the correspondence may have errors? Will other some other 2D-based self-supervision-based method for correspondence learning for an image with different transformations could also be used to assist the proposed method? Can the author have a short discussions about them with the proposed method. For some examples, there are 2D-based self-supervised learning work for co-part segmentation and part correspondence.

Co-part segmentation:
1. Hung, Wei-Chih, Varun Jampani, Sifei Liu, Pavlo Molchanov, Ming-Hsuan Yang, and Jan Kautz. "Scops: Self-supervised co-part segmentation." In Proceedings of the IEEE/CVF Conference on Computer Vision and Pattern Recognition, pp. 869-878. 2019.
2. Ziegler, Adrian, and Yuki M. Asano. "Self-Supervised Learning of Object Parts for Semantic Segmentation." In Proceedings of the IEEE/CVF Conference on Computer Vision and Pattern Recognition, pp. 14502-14511. 2022.

Semantic correspondence:
1. Novotny, David, Samuel Albanie, Diane Larlus, and Andrea Vedaldi. "Self-supervised learning of geometrically stable features through probabilistic introspection." In Proceedings of the IEEE Conference on Computer Vision and Pattern Recognition, pp. 3637-3645. 2018.

**Limitations:**

Yes, the authors do address the limitations of the proposed method and potential negative societal impact of their work at the end of the main paper.

**Strengths And Weaknesses:**

Strength:

1. The work shows that by performing metric learning for the local representation of an object across multiple view could significantly improve the low-shot classification of novel categories.
2. They conduct the experiments on both synthetic and real datasets and achieve consistent improvement for different scenarios.

Weakness:

1. The evaluation part is insufficient. The compared methods are all before 2021, and there is no 2022 method for comparison, and only two self-supervised baseline method for comparison, VISPE and VISPE++, which are proposed in 2020.
2. The training heavily relies on the accuracy of the camera parameter estimator, COLMAP and foreground segmentation method, PointRend. There is not much discussion when the preprocessing results are noisy.

---

> ### Author Response · Authors · 2022-08-02
> **To Reviewer 3MZa - Answers to Questions continued**
>
> **R2 Q5**: *Is the 3D-based self-supervision necessary since the correspondence may have errors? Will other some other 2D-based self-supervision-based method for correspondence learning for an image with different transformations could also be used to assist the proposed method? Can the author have a short discussions about them with the proposed method. For some examples, there are 2D-based self-supervised learning work for co-part segmentation and part correspondence.*
>
> Co-part segmentation:
>
> [A] Hung, Wei-Chih, Varun Jampani, Sifei Liu, Pavlo Molchanov, Ming-Hsuan Yang, and Jan Kautz. "Scops: Self-supervised co-part segmentation." In Proceedings of the IEEE/CVF Conference on Computer Vision and Pattern Recognition, pp. 869-878. 2019.
>
> [B] Ziegler, Adrian, and Yuki M. Asano. "Self-Supervised Learning of Object Parts for Semantic Segmentation." In Proceedings of the IEEE/CVF Conference on Computer Vision and Pattern Recognition, pp. 14502-14511. 2022.
>
> Semantic correspondence:
>
> [C] Novotny, David, Samuel Albanie, Diane Larlus, and Andrea Vedaldi. "Self-supervised learning of geometrically stable features through probabilistic introspection." In Proceedings of the IEEE Conference on Computer Vision and Pattern Recognition, pp. 3637-3645. 2018.
>
> **R2 A5:**
> We thank the reviewer for raising this and present a discussion of our work relative to the works the reviewer mentions. We also present an empirical study demonstrating the value of 3D-based multi-view self supervision. We will add these papers to our related work, and add an additional discussion.
>
> *Discussion Comparing with  [A]*
>
> SCOPS is a method for self supervised object part co-segmentation of single categories, however it is designed for fine grained scenarios (faces, birds, and single categories of PASCAL VOC). Learning fine-grained part co-segmentation for one visual category is a different problem from our setting, where our goal is self-supervised learning of a dense, local feature representation for many object categories at once.
>
> For example, SCOPS requires the number of parts to be pre-defined (they use between 4 and 8) for its semantic consistency loss. The ability to pre-define a number of parts and apply a semantic consistency loss is only feasible in fine grained settings where a well defined parts taxonomy exists that is shared across all objects being learned (eyes, nose, mouth for faces or wings, body, head, tail for birds). This is not possible for our setting. Further, the equivariance loss they propose is between two 2D geometric transforms of an image, whereas our contrastive loss can be thought of as an equivariance loss between two different 3D viewpoints of an object. Please see our empirical study in **R4 Q4** that shows that this is insufficient for low-shot generalization.
>
> *Discussion comparing with [C]*
>
> Novotny et al. [C] propose to learn dense visual descriptors using 2D geometric and color transformations of images from multiple categories at once. They apply a probabilistic loss to learn how to encode the corresponding pixels of the two transformed images using the same features. This probabilistic loss can be applied between two different 3D viewpoints of an object, but it would require the same geometric information to derive correspondences as our approach.
>
> *Discussion comparing with [B]*
>
> Leopart is a visual transformer-based method for self-supervised pre-training of representations for semantic segmentation. It falls within the family of dense self-supervised learning works like VaDeR [D], DenseCL [E] and PixPro [F]. It uses a clustering based strategy to learn from different crops of one image. A clustering-based approach can be applied in a multi-view setting rather than a contrastive one, but the optimal cluster assignments would still require 3D information.
> Leopart belongs to the family of 2D dense self-supervised. Please refer to *R4 Q4* for discussion regarding comparisons to 2D dense contrastive learning.
>
> [D] Pedro O O Pinheiro, Amjad Almahairi, Ryan Benmalek, Florian Golemo, and Aaron C Courville. Unsupervised learning of dense visual representations. NeurIPS 2020.
>
> [E] Wang, Xinlong, et al. "Dense contrastive learning for self-supervised visual pre-training." Proceedings of the IEEE/CVF Conference on Computer Vision and Pattern Recognition. 2021.
>
> [F] Xie, Zhenda, et al. "Propagate yourself: Exploring pixel-level consistency for unsupervised visual representation learning." Proceedings of the IEEE/CVF Conference on Computer Vision and Pattern Recognition. 2021.

---

> ### Author Response · Authors · 2022-08-02
> **To Reviewer 3MZa - Questions**
>
> **R2 Q3:** *How are the views sampled for the training? If the object views are too extreme, the estimation of COLMAP may fail.*
>
> **R2 A3:** We directly use the [CO3D dataset](https://arxiv.org/pdf/2109.00512.pdf) videos and COLMAP outputs released by the CO3D authors. The dataset consists of crowdsourced videos taken by walking around objects in indoor and outdoor settings, resulting in an overall diverse set of views. (We provide some visualizations [at this link](https://imgur.com/a/iFyzlEy)). Given that DOPE can train and obtain competitive quantitative results, we believe that COLMAP is sufficiently robust.
>
> ---
>
> **R2 Q4:** *How do the preprocessing errors from the camera parameters, sparse depth, and segmentation map affect the performance? Although w in appendix, the authors do provide the ablation studies of w/o random background remove and w/o mask prediction.*
>
> **R2 A4:**  Based on this suggestion we performed an empirical investigation into the robustness of DOPE to noisy segmentation masks. We obtained the segmentation masks using an off the shelf unsupervised instance segmentation method, freeSOLO [A]. (We did this for ABC under the time pressure of the rebuttal by directly applying pre-trained freeSOLO on our data domain without additional training. In the final version we will repeat this for our synthetic data and CO3D-LS with a finetuned freeSOLO which will yield better performance).
>
> We present results for training for DOPE and VISPE++ on ABC with ground truth (GT) masks and freeSOLO predicted masks, and testing low-shot generalization on ModelNet. The category supervised RFS is provided as reference. We observe that while performance decreases for DOPE when it is trained with freeSOLO masks as opposed to GT masks, the performance is still competitive with category supervised methods. Note that there is a domain gap between our data and the freeSOLO training data and that training freeSOLO on our data domains for mask estimation will lead to improved performance.
>
> | | 1-shot 5-way | 5-shot 5-way| 1-shot 10-way | 5-shot 10-way |
> | --| ----------- | ----------- | ------------- | ----------------|
> RFS - Category Supervised| 50.85 (0.40) | 66.48 (0.35) | 37.11 (0.33) | 53.37 (0.16) |
> VISPE++ -- ABC with GT mask |  58.31 (0.40) | 73.40 (0.15) | 44.91 (0.15) | 61.40 (0.08) |
> VISPE++ -- ABC with freeSOLO mask | 50.28 (0.31) | 64.95 (0.33) | 36.80 (0.22) | 51.37 (0.18) |
> DOPE -- ABC with GT mask | 56.82 (0.40) | 72.61 (0.31) | 44.02 (0.25) | 61.22 (0.17) |
> DOPE -- ABC with freeSOLO mask | 52.34 (0.40) | 66.17 (0.32) | 38.87 (0.22) | 52.78 (0.18) |
>
> [A] Wang, Xinlong, et al. "FreeSOLO: Learning to Segment Objects without Annotations." Proceedings of the IEEE/CVF Conference on Computer Vision and Pattern Recognition. 2022.

---

> ### Author Response · Authors · 2022-08-02
> **To Reviewer 3MZa - Answers to Weaknesses**
>
> **R2 Q1:** *The evaluation part is insufficient. The compared methods are all before 2021, and there is no 2022 method for comparison, and only two self-supervised baseline method for comparison, VISPE and VISPE++, which are proposed in 2020.*
>
> **R2 A1:** We would like to emphasize that regarding the supervised baselines, in our paper we outperform SupMoCo which is from 2021 and is state of the art on MetaDataset. Although one of our other baselines, FEAT [A] is from CVPR 2020 it is still a strong baseline because it outperforms more recently published works on MiniImagenet and TieredImagenet like MetaBaseline [B].
>
> However, in response to the reviewer’s comment we conducted an additional experiment using a more recent self-supervised learning baseline algorithm. We evaluate against SimSiam-VISPE++, which is based on the more recent self-supervised method SimSiam [C] from CVPR 2021 rather than the MoCoV2-VISPE++ in the original submission. This model was trained with the same setting as VISPE++ in our main text, where positives come from two images of different viewpoints from the same object using the same data augmentation strategy as our method and VISPE++. The multi-view training of DOPE utilizing local information maintains its advantage on the ModelNet40-LS test set.
>
> Low-Shot Category Generalization on the ModelNet40-LS Test Set
> | | 1-shot 5-way | 5-shot 5-way| 1-shot 10-way | 5-shot 10-way |
> | --| ----------- | ----------- | ------------- | ----------------|
> MoCov2-VISPE++ - ABC | 56.50 (0.43) | 69.75 (0.32) | 42.97 (0.24) | 56.90 (0.22)
> SimSiam-Vispe++ - ABC | 56.10 (0.42) | 62.17 (0.39) |  43.16 (0.24) | 49.44 (0.22)
> DOPE - ABC | 57.50 (0.42)  | 71.56 (0.31) | 44.14 (0.23) | 59.14 (0.23)
>
> [A] Ye, Han-Jia, et al. "Few-shot learning via embedding adaptation with set-to-set functions." Proceedings of the IEEE/CVF Conference on Computer Vision and Pattern Recognition. 2020.
>
> [B] Chen, Yinbo, et al. "Meta-baseline: Exploring simple meta-learning for few-shot learning." Proceedings of the IEEE/CVF International Conference on Computer Vision. 2021.
>
> [C] Chen, Xinlei, and Kaiming He. "Exploring simple siamese representation learning." Proceedings of the IEEE/CVF Conference on Computer Vision and Pattern Recognition. 2021.
>
> ---
>
> **R2 Q2:**  *The training heavily relies on the accuracy of the camera parameter estimator, COLMAP and foreground segmentation method, PointRend. There is not much discussion when the preprocessing results are noisy.*
>
> **R2 A2**: Despite the difficulty and noise inherent in the CO3D dataset (please see [this link](https://imgur.com/a/iFyzlEy) for visualization of the data), DOPE can still obtain competitive performance when trained on this data. Please refer to our answer to **R2 Q4** for an empirical investigation into the effects of foreground segmentation noise.

---

### Official Review · Reviewer_7WGn · 2022-07-12

**Rating:** 5
**Confidence:** 3
**Soundness:** 3 good
**Presentation:** 3 good
**Contribution:** 3 good

**Summary:**

This paper aims to learn representative features from multi-view images of objects without requiring category labels. The authors propose a Deep Object Part Encodings (DOPE) framework that leverages sparse depths, foreground masks and camera poses to obtain pixel-level corresponds between different views and performs self-supervised learning from these views. DOPE works well on low-shot classification of novel categories.

**Questions:**

Please refer to weakness. My major concern is the foreground mask is not always available. It is necessary to show DOPE can work well with unsupervised foreground masking methods.

Update:
A minor update of review comments:
I think it is fine that the author can also leverage class-agnostic segmentation models or segmentation models pre-trained on other classes instead of unsupervised methods. In other words, any segmentation models that are NOT trained on the target categories will be fine for me. Please feel free to reply this question.

**Strengths And Weaknesses:**

Strengths:
1) The overall idea is interesting and novel. Compared to previous work [1], this paper explicitly leverages the information of 3D geometry in multi-view images for self-supervised learning.
2) The performance is good. DOPE outperforms other counterparts and performs competitively with the supervised methods.
3) The paper is well written.

Weaknesses:

The key weakness is that DOPE relies on ground-truth foreground masks, which will limit its application. It is necessary to explore using unsupervised foreground masking methods (e.g., [2]) to extract masks and show if DOPE can still work well.



[1] Exploit clues from views: Self-supervised and regularized learning for multiview object recognition.

[2] Unsupervised Foreground Extraction via Deep Region Competition

---

> ### Author Response · Authors · 2022-08-02
> **To Reviewer 7WGn - Answers to Weaknesses and Questions**
>
> **R1 Q1:** *The key weakness is that DOPE relies on ground-truth foreground masks, which will limit its application. It is necessary to explore using unsupervised foreground masking methods (e.g., [A]) to extract masks and show if DOPE can still work well.*
>
> [A] Unsupervised Foreground Extraction via Deep Region Competition
>
> **R1 A1**: Motivated by this reviewer's comment, we perform an investigation into whether DOPE still works well when unsupervised foreground segmentation methods are used. We use the more recent freeSOLO [B] from CVPR 2022, trained without any category or segmentation labels on ImageNet and COCO. (We did this for ABC under the time pressure of the rebuttal by directly applying pre-trained freeSOLO on our data domain without additional training. In the final version we will repeat this for our synthetic data and CO3D-LS with a properly trained freeSOLO which will yield better performance).
>
> We present results for training for DOPE and VISPE++ on ABC with ground truth (GT) masks and freeSOLO predicted masks, and testing low-shot generalization on ModelNet. The category supervised RFS is provided as reference. We observe that performance decreases for DOPE when it is trained with freeSOLO masks as opposed to GT masks, but conclude that DOPE still works well as the performance is still competitive with the category-supervised RFS and VISPE++. Note that there is a domain gap between our data and the freeSOLO training data and that training freeSOLO on our data domains for mask estimation will lead to improved performance.
>
> | | 1-shot 5-way | 5-shot 5-way| 1-shot 10-way | 5-shot 10-way |
> | --| ----------- | ----------- | ------------- | ----------------|
> RFS - Category Supervised| 50.85 (0.40) | 66.48 (0.35) | 37.11 (0.33) | 53.37 (0.16) |
> VISPE++ -- ABC with GT mask |  58.31 (0.40) | 73.40 (0.15) | 44.91 (0.15) | 61.40 (0.08) |
> VISPE++ -- ABC with freeSOLO mask | 50.28 (0.31) | 64.95 (0.33) | 36.80 (0.22) | 51.37 (0.18) |
> DOPE -- ABC with GT mask | 56.82 (0.40) | 72.61 (0.31) | 44.02 (0.25) | 61.22 (0.17) |
> DOPE -- ABC with freeSOLO mask | 52.34 (0.40) | 66.17 (0.32) | 38.87 (0.22) | 52.78 (0.18) |
>
> [B] Wang, Xinlong, et al. "FreeSOLO: Learning to Segment Objects without Annotations." Proceedings of the IEEE/CVF Conference on Computer Vision and Pattern Recognition. 2022.
>
> ---
>
> **R1 Q2:**  *A minor update of review comments: I think it is fine that the author can also leverage class-agnostic segmentation models or segmentation models pre-trained on other classes instead of unsupervised methods. In other words, any segmentation models that are NOT trained on the target categories will be fine for me. Please feel free to reply this question.*
>
> **R1 A2:** We agree that obtaining masks from segmentation models trained on the target categories limits the potential applicability of DOPE. In our answer **R1 A1** we provide a direct investigation for a fully unsupervised instance segmentation model.

---

> > ### Comment · Reviewer_7WGn · 2022-08-09
> > **My Final Rating**
> >
> > The rebuttal solves my concerns well. I keep my previous rating to borderline accept this paper.

---

### Author Response · Authors · 2022-08-02
**To all reviewers**

We thank the reviewers for their detailed and thoughtful feedback. We are glad they found our work to be interesting and novel, well written, with consistent improvement on both synthetic and real datasets. Based on the reviewers’ suggestions we have conducted four additional experiments reported in this rebuttal to shed more light on our approach.

**Contribution and Novelty:** The central theme of this work is to compare two sources of information for learning few shot categorization: 1) Category labels and single views, vs 2) Multiple views with additional geometric information (and no category labels). We believe it is surprising that our purely geometric training approach is so effective for few shot categorization, surpassing category-trained benchmarks in the synthetic data case and exhibiting on-par performance on real data. Moreover, our competitive real world results are obtained using noisy estimated geometric information (camera calibration and sparse depth). In the broader context of self-supervised learning, we have gone farther than any prior few shot learning work in demonstrating the power of geometric supervision to overcome the need for category labels during training, and we believe this is an important step for the field.

---

### Meta-Review · Area_Chair_uPyw · 2022-08-24

**Recommendation:** Accept
**Confidence:** Certain

**Metareview:**

**Summary**: This paper aims to learn representative features of objects (without category labels) from multi-view images. A Deep Object Part Encodings (DOPE) framework is proposed, which leverages sparse depths, foreground masks, and camera poses (from COLMAP) to obtain correspondence across different views and performs self-supervised learning from these views. DOPE works well on the low-shot classification of novel categories.

**Strengths**: The overall idea is interesting, novel, and effective. It explicitly leverages 3D geometry in multi-view images for self-supervised learning. The performance gain is notable. The experimental design is solid. The paper is generally well written.

**Weaknesses**: DOPE relies on ground-truth foreground masks and the estimated camera poses by COLMAP. The compared baselines are not sufficient (mostly before 2021). Some technical parts and claims/definitions are not clear.

**Recommendation**: The paper receives positive ratings and reviews in general. After rebuttal, most of the reviewers’ concerns are addressed (e.g., additional experiments) and the paper clearly has strengths. The AC thus suggests acceptance. The AC strongly suggests that the authors incorporate their rebuttal (e.g., responses to Reviewer 3MZa; change the title) into their camera-ready version.

**Award:**

No

---

### Decision · Program_Chairs · 2022-09-14

Accept